# Human-like monocular depth biases in deep neural networks

**Yuki Kubota** *, **Taiki Fukiage**

Communication Science Laboratories, NTT, Inc., Kanagawa, Japan

* yuki.kubota95@gmail.com

**Data availability statement:** All human and model data, code for generating model

## Abstract

Human depth perception from 2D images is systematically distorted, yet the nature of these distortions is not fully understood. By examining error patterns in depth estimation for both humans and deep neural networks (DNNs), which have shown remarkable abilities in monocular depth estimation, we can gain insights into constructing functional models of this human 3D vision and designing artificial models with improved interpretability. Here, we propose a comprehensive human-DNN comparison framework for a monocular depth judgment task. Using a novel human-annotated dataset of natural indoor scenes and a systematic analysis of absolute depth judgments, we investigate error patterns in both humans and DNNs. Employing exponential-affine fitting, we decompose depth estimation errors into depth compression, per-image affine transformations (including scaling, shearing, and translation), and residual errors. Our analysis reveals that human depth judgments exhibit systematic and consistent biases, including depth compression, a vertical bias (perceiving objects in the lower visual field as closer), and consistent per-image affine distortions across participants. Intriguingly, we find that DNNs with higher accuracy partially recapitulate these human biases, demonstrating greater similarity in affine parameters and residual error patterns. This suggests that these seemingly suboptimal human biases may reflect efficient, ecologically adapted strategies for depth inference from inherently ambiguous monocular images. However, while DNNs capture metric-level residual error patterns similar to humans, they fail to reproduce human-level accuracy in ordinal depth perception within the affine-invariant space. These findings underscore the importance of evaluating error patterns beyond raw accuracy, providing new insights into how humans and computational models resolve depth ambiguity. Our dataset and methodology provide a framework for evaluating the alignment between computational models and human perceptual biases, thereby advancing our understanding of visual space representation and guiding the development of models that more faithfully capture human depth perception.

predictions, data-collection and data-analysis scripts, and a usage protocol are publicly available from https://osf.io/apkqn/.

**Funding:** This work was supported by JSPS KAKENHI (Grant Number 23KJ2226 to YK). The funders had no role in the research design, data collection and analysis, decision to publish, or preparation of the manuscript.

**Competing interests:** The authors have declared that no competing interests exist.

## Author summary

Understanding the characteristics of errors in depth judgments exhibited by humans and deep neural networks (DNNs) provides a foundation for developing functional models of human brain and artificial models with enhanced interpretability. To address this, we constructed a human depth judgment dataset using indoor photographs and compared human depth judgments with those of DNNs. Our results show that humans systematically compress far distances and exhibit distortions related to viewpoint shift, which remain remarkably consistent across observers. Strikingly, the better the DNNs were at depth estimation, the more they also exhibited human-like biases. This suggests that these seemingly suboptimal human biases could in fact reflect efficient strategies for inferring 3D structure from ambiguous 2D inputs. However, we also found that the models' error patterns are more similar to each other than to humans, and they struggle to match human performance in judging the relative order of objects in depth, especially when we accounted for viewpoint distortions. We believe that our dataset and discovery of multiple error factors will drive further comparative studies between humans and DNNs, facilitating model evaluations that go beyond simple accuracy to uncover how depth perception truly works—and how it might best be replicated in computational models.

## 1. Introduction

How does the human visual system transform the inherently flat and ambiguous images on the retina into a coherent three-dimensional perception of space? This question has long been a central focus in vision science. While a single image provides multiple cues—from subtle variations in texture and shading to the geometric regularities of perspective—the transformation from two to three dimensions is fundamentally underdetermined. In fact, our perception of depth from monocular cues is demonstrably not a simple, accurate read-out of the external world. Instead, it is shaped by a range of systematic distortions, resulting in a perceived space that deviates in complex ways from Euclidean geometry [1]. These biases encompass a variety of phenomena, including compressed depth perception, where distances at greater depths are systematically underestimated [2–8]; shifts in vanishing point, which alters the overall geometry of the scene [9,10]; characteristic distortions observed when viewing photographs [11,12]; and context-dependent depth perception, with environmental factors modulating how distance cues are weighted [13–16]. The precise nature of these distortions, their underlying causes within the visual system, and the form of the spatial representation that emerges from monocular vision remain open and actively debated questions. Clarifying these issues is crucial for understanding how we perceive and interact with the three-dimensional world around us.

One promising avenue for making progress on these questions lies in comparing human depth perception with artificial systems. In recent years, deep neural networks (DNNs) have emerged as powerful computational tools that can achieve impressive performance in monocular depth estimation [17,18]. These models, trained on vast datasets of images, offer a unique opportunity to investigate the complexities of depth perception from a computational perspective. By creating artificial systems that grapple with the same ill-posed problem faced by the human visual system, we can explore potential solutions and, crucially, compare their behavior to human perception. Do DNNs, in their pursuit of accurate depth estimation,

converge on solutions that resemble human strategies? Or do they exploit fundamentally different computational pathways, potentially sidestepping the very distortions that characterize human spatial vision?

These questions about the alignment of human and DNN strategies have been actively explored in other areas of vision research. While this is particularly notable in high-level tasks such as object recognition—a function often associated with the ventral stream of the human brain—such comparisons now extend to a wide range of domains. These include mid-level perceptual judgments on material properties [19–21], as well as other fundamental cognitive functions like visual attention [22], and working-memory [23]. In object recognition, early studies, such as those comparing CNN representations to activity in the primate inferotemporal cortex (IT) [24], suggested intriguing parallels, fueling initial excitement that DNNs might serve as effective models of biological visual processing. Indeed, DNNs have achieved impressive, even human-level, performance on object recognition benchmarks like ImageNet [25]. However, subsequent, more nuanced comparisons have revealed a complex picture. Counterintuitively, some research has shown a divergence between increasing ImageNet accuracy and alignment with human neural and behavioral responses [26,27]. These studies demonstrate that simply achieving high accuracy on a benchmark does not guarantee human-like perception. Instead, DNNs often exhibit qualitatively different error patterns from humans, rely on different visual cues (e.g., an over-reliance on texture rather than shape) [28], and are vulnerable to what are known as adversarial examples—subtly modified images that are imperceptible to humans but can completely fool DNNs, causing them to make nonsensical classifications [29]. In response to these findings, researchers have increasingly emphasized the importance of moving beyond simple performance metrics and examining the similarity of error patterns between humans and DNNs as a more stringent test of model fidelity to biological vision [30,31].

While in-depth comparisons between humans and DNNs are becoming increasingly common for object recognition and ventral stream functions, a small but growing number of efforts have begun to investigate mid-level vision and dorsal stream processes. For example, Yang et al. [32] used psychophysical methods to map human-perceived motion flow in naturalistic movies, finding that while DNN models shared some lower-level errors with human perception, they often failed to reproduce the systematic "flow illusions" driven by higher-level mechanisms. For the depth estimation task, existing studies mainly focus on visualizing DNN inference for depth [33], exploring correlations between model characteristics such as shape bias and accuracy [34], or improving the interpretability of depth prediction [35]. Danier et al. [36] also proposed a benchmark to evaluate how vision models utilize various monocular depth cues, although they did not include comparisons with human depth judgments. A few other works have tested specific artificial images [37,38], and one study compared relative depth judgments between humans and a limited set of models [39]. While these initial findings offer useful insights, a broader exploration covering diverse DNN architectures and training methods—particularly comparing systematic biases and error patterns between humans and DNNs—remains an open and critical area of investigation.

To directly address this critical need for systematic human-DNN comparisons in depth judgment, our study aims to explore behavioral similarities between humans and DNNs in this domain. A key challenge in pursuing this objective is the current lack of effective datasets specifically designed for monocular depth judgment comparisons. While valuable human-annotated datasets or collection methodologies exist—such as those focusing on ordinal depth relationships and surface normals [40–42], interactive sketching [43–46], and image ranking techniques [47]—these resources often present limitations for our comparative goals. Specifically, they frequently treat human judgments as a form of ground truth, potentially

overlooking the valuable information contained in the inherent uncertainties and biases of human perception. Furthermore, the absence of human-annotated datasets directly linking RGB-D images with physical depth data makes it difficult to quantitatively evaluate the deviations between human and DNN judgments relative to a shared, objective ground truth.

Therefore, in this study, we collected human depth judgments using images from the NYU Depth V2 dataset [48], a widely recognized benchmark for indoor monocular depth estimation. Recognizing the impracticality of pixel-wise human depth annotations, we focused on collecting absolute depth estimates for a curated set of points within each image (Fig 1, the leftmost column). This strategy ensured both the feasibility of psychophysical data collection and direct comparability with DNN outputs at corresponding image locations. To quantitatively analyze the error patterns in both human and DNN depth judgments, we employed exponential-affine fitting. This method is grounded in psychophysical findings that human depth perception is subject to both depth compression [2–8] and affine distortions [49–53]. Exponential fitting captures the non-linear compression of perceived depth with increasing distance, while affine fitting accounts for linear transformations, including scale, shift, and shear, allowing us to decompose systematic biases beyond simple scaling errors (Fig 1, column labeled "Decomposing Errors").

Applying this framework to our collected data, we reveal systematic biases in human depth judgments, characterized by depth compression and affine distortions. Intriguingly, we demonstrate that DNNs trained for monocular depth estimation also partially exhibit these

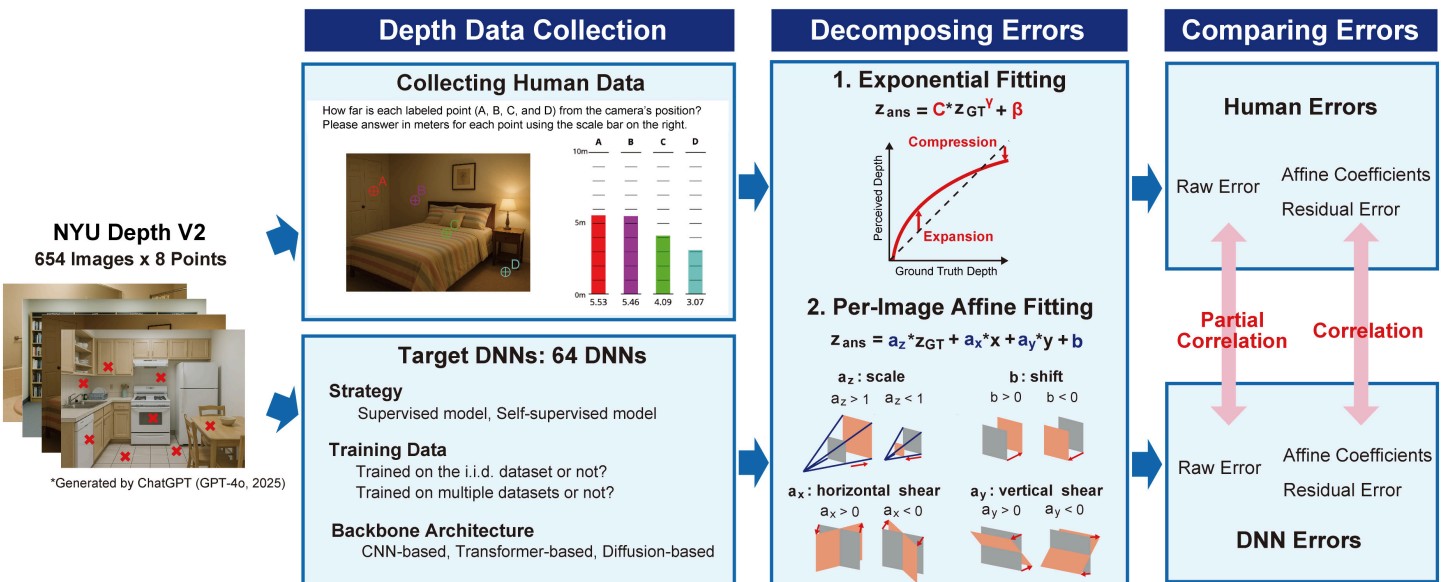

**Fig 1. Overview of our work.** (Left) Depth Data Collection: We utilized the NYU Depth V2 dataset, containing 654 test images of indoor scenes, and selected eight target points per image (example images with red crosses). For collecting human data, participants provided absolute depth judgments (in meters) for each labeled point (A, B, C, D) by adjusting bars against reference lines, as shown in the example task screen. We evaluated a diverse set of 64 DNN models, varying in strategy (supervised, self-supervised, hybrid, generative), training data characteristics (i.i.d. vs. o.o.d. datasets, single vs. multiple datasets), and backbone architecture (CNN-based, Transformer-based, Diffusion-based). The image was generated by ChatGPT (GPT-4o, 2025) to avoid potential copyright issues and is intended for illustrative purposes only. (Center) Decomposing Errors: We decomposed errors in depth judgments using two fitting methods. *1. Exponential Fitting* was applied to capture the global depth compression effect at far range as well as expansion at near range in perceived depth. *2. Per-Image Affine Fitting* was performed to analyze per-image affine transformations. This model decomposes errors into scale, shift, horizontal shear, and vertical shear components, schematically depicted with orange shapes transforming from ground truth (gray shapes). (Right) Comparing Errors: We compared error patterns between humans and DNNs by calculating partial correlations between raw errors and correlations between affine coefficients and residual errors, allowing for a quantitative assessment of human-DNN similarity in depth judgments.

human-like biases, particularly depth compression, and that higher accuracy in DNNs correlates with greater similarity to human error patterns in terms of affine components. These findings establish a foundation for advancing comparative studies in depth estimation and suggest the potential for developing more human-like depth estimation models in a data-driven manner. Moreover, aligning these mid-level visual representations with human-like depth perception could offer substantial advantages for higher-level applications such as robotic navigation and autonomous systems [54,55].

In the following sections, we first present a comprehensive characterization of human depth biases through exponential-affine fitting. We then systematically compare these biases with those of a diverse range of DNN models, analyzing similarities and differences at both the metric level and the ordinal level.

## 2. Results

### 2.1. Dataset collection for human depth judgments

To compare human and DNN depth judgments under the same conditions, we conducted psychophysical experiments using a depth judgment task. The stimuli consisted of 654 test images from the NYU Depth V2 dataset [48], which features natural indoor photographs. We selected the NYU dataset for depth judgment evaluation for two primary reasons. First, natural photographs are familiar to both humans and DNNs. In contrast, controlled artificial images are typically out of distribution (o.o.d.) for most DNNs trained primarily on natural images. Using artificial images could confound the analysis by making it difficult to discern whether any observed errors in DNNs stemmed from model-specific biases or from mismatched image statistics. Second, the NYU dataset includes a wide variety of 3D structures such as walls, floors, ceilings, and various objects. We hypothesized that the depth structures in these images are more complex than those in outdoor scenes, which are often uniformly captured by in-vehicle cameras. This increased complexity has the potential to reveal non-trivial patterns in depth judgments made by both humans and DNNs.

To obtain absolute depth estimates, we asked participants to report distances (in meters) for specific points within each image (Fig 1, the column labeled "Depth Data Collection"). Requesting depth estimates for every pixel would have been impractical; therefore, the task was limited to the preselected target points. For each of the 654 images, eight specific locations were randomly selected as targets for depth judgments. To minimize potential ambiguities, points near segmentation boundaries or at the edges of the image were excluded from selection. Consequently, we prepared stimuli consisting of 654 test images, each containing eight designated target points for absolute depth judgments. To further reduce cognitive load, the eight targets per image were divided into two groups of four, with the first group shown in the first trial and the second group in the subsequent trial.

Although previous studies have predominantly employed relative depth judgment tasks, where participants assess pairwise depth relationships [13,40,42,50,52], our study required absolute depth judgments. This approach allowed us to quantitatively assess both the uncertainty and systematic (quantitative) biases inherent in human depth judgments—factors that are not fully captured by pairwise comparisons alone. By having participants evaluate four points simultaneously, we aimed to leverage implicit pairwise comparisons while also encouraging them to consider both the local and overall depth structures in the image. Additionally, to evaluate how task type might affect results, we collected a supplemental dataset based on relative depth judgments (two-alternative forced choice). Details of this additional dataset are provided in Materials and methods.

Human depth judgments were collected through a crowd-sourced online survey involving 898 participants aged between their 20s and 40s. The experiment was designed to be completed within approximately 30 minutes so that participants could complete it comfortably while staying attentive. Following data collection, participants whose responses deviated significantly from the group average were excluded as outliers, resulting in 742 valid participants out of 898. The average (minimum and maximum) number of responses for each data point after screening was 18.94 (14 – 25).

To establish a human-level baseline or "ceiling" for evaluating the performance of the DNNs, we needed to quantify the consistency of depth judgments across human observers. In the experiment, each participant responded to only a subset of the 654 images. To reliably measure how humans agree with each other across the full dataset, we adopted a random half-split procedure. This involves creating two "pseudo-participant" datasets by randomly splitting and averaging the observers. This approach provides full data coverage and, by averaging out individual-specific response noise, allows for a more robust measurement of the error patterns that are systematically shared among human observers. By repeating this split 1,000 times, we obtained robust estimates and their 95% confidence intervals (CIs) for all subsequent human data analyses unless otherwise specified (see Materials and methods for detail).

## 2.2. Common biases in depth judgments among humans

### 2.2.1. Human depth judgments exhibit a substantially shared error pattern.

We began our analysis by examining how closely participants' depth judgments aligned with the ground truth. Fig 2 illustrates the relationship between the ground truth and the averaged depth judgments. Using the random half-split approach described earlier, the expected root mean squared error (RMSE) between the pseudo-participant data and the ground truth was 1.261 (95%CI [1.250,1.274]). We also observed an expected correlation of 0.674 (95%CI [0.656,0.691]) between the ground truth and these human judgments. Next, we evaluated consistency across participants. The inter-individual Pearson correlation between two separate random half-splits of the dataset was 0.895 (95%CI [0.871,0.916]), indicating a high level of agreement in depth estimates. To more precisely assess similarities in participants' error patterns, we computed a Pearson partial correlation between the split datasets while controlling for the ground truth. This approach allows us to distinguish whether the high consistency between observers arises simply because they are both accurately perceiving the true depth, or because they share a common pattern of systematic biases. Specifically, let $\mathbf{z}_1$ and $\mathbf{z}_2$ be the first and second split dataset, and $\mathbf{z}_{\mathrm{GT}}$ be the ground truth depths. Pearson partial correlation measures the linear relationship between $\mathbf{z}_1$ and $\mathbf{z}_2$ after statistically removing the effect of $\mathbf{z}_{\mathrm{GT}}$. Formally,

$$r_{\mathbf{z}_1,\mathbf{z}_2|\mathbf{z}_{\mathrm{GT}}} = \frac{r_{\mathbf{z}_1,\mathbf{z}_2} - r_{\mathbf{z}_1,\mathbf{z}_{\mathrm{GT}}}\, r_{\mathbf{z}_2,\mathbf{z}_{\mathrm{GT}}}}{\sqrt{\left(1 - r^2_{\mathbf{z}_1,\mathbf{z}_{\mathrm{GT}}}\right)\left(1 - r^2_{\mathbf{z}_2,\mathbf{z}_{\mathrm{GT}}}\right)}},$$

where $r_{\mathbf{z}_1,\mathbf{z}_2}$ is the Pearson correlation between $\mathbf{z}_1$ and $\mathbf{z}_2$, and similarly for the other terms. This analysis yielded an expected partial correlation of 0.808 (95%CI [0.766,0.845]), suggesting a strong common bias pattern across observers that is not simply explained by the ground truth alone. Notably, even when using the unscreened dataset, we observed a correlation of 0.896 (95%CI [0.871,0.917]) and a partial correlation of 0.808 (95%CI [0.766,0.846]), alleviating concerns that the screening procedure might artificially inflate these results. Furthermore, a similar analysis performed on the original raw data (i.e., before the half-split averaging)

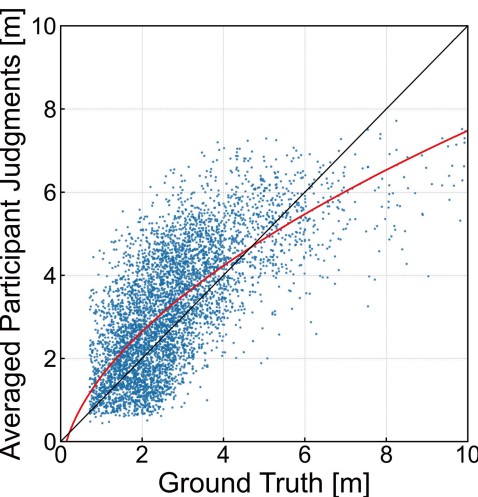

**Fig 2. Scatter plot of averaged human depth judgments for each data point.** The red curve represents the exponential fitting results.

also revealed a strong positive correlation among observers: an average correlation of 0.669 (95%CI [0.659,0.679]) and a partial correlation of 0.508 (95%CI [0.495,0.521]), confirming that this high consistency is not an artifact of our procedure.

**2.2.2. Human depth judgments exhibit depth compression effect.**   Fig 2 reveals that as the distance increases, perceived depth becomes progressively more compressed. This observation aligns with the depth compression effect documented in previous studies [5,6]. To quantitatively analyze this effect, we adopted exponential fitting:

$$z_{\text{ans}} \simeq C * z_{\text{GT}}^{\gamma} + \beta, \tag{1}$$

where $C$ controls the overall scaling, $\gamma$ determines how nonlinearly perceived depth changes with actual distance, and $\beta$ accounts for any systematic shift. We performed this fitting on "pseudo-participant data" (generated by randomly sampling half of the participants and averaging their responses) 1,000 times to estimate the expected fitting parameters and their 95% confidence intervals.

As a result, the coefficients of exponential fitting were determined as follows: $C : 2.37$ (95%CI [2.36,2.40]), $\gamma : 0.543$ (95%CI [0.542,0.545]), and $\beta : -0.808$ (95%CI [−0.835,−0.795]). The exponent $\gamma$ primarily influences the shape of the fitting curve at greater distances, where its effect on depth compression is more pronounced, while the scale $C$ has a stronger impact on the fitted values at closer distances. Consequently, the results of $\gamma < 1.0$ and $C > 1.0$ indicate a compression effect for far distances and an expansion effect for near distances. These trends are consistent not only with the depth compression effect reported in the previous studies [5,6], but also with the depth expansion effect observed in photographs [12].

**2.2.3. Human depth judgments exhibit consistent affine biases.**   In addition to the depth compression effect, what other factors contribute to systematic biases in human depth judgment data? Here, we employ the affine model as an analytical framework [49–53]. This framework, inspired by the idea that human depth perception from 2D images may process information invariant to affine transformations, allows us to decompose systematic biases in depth

judgments into interpretable components. The affine model posits that our perceptual system implicitly accounts for affine distortions—transformations encompassing scaling, shearing, and translation—which are inherent in the projection of a 3D world onto a 2D image, and also in potential systematic distortions within the visual system. This framework suggests that to achieve stable and robust depth perception despite variations in viewpoint and projection geometry, the visual system might extract and utilize affine invariant representations from visual input. In essence, the affine model recognizes that when we perceive depth from images, we may not be directly and veridically recovering Euclidean depth, but rather operating within a perceptual space that may be systematically related to, but not identical to, the true metric space. Going beyond simple scaling and shifting ambiguities, this model incorporates global vertical and horizontal shearing—transformations that can arise from variations in viewpoint. By fitting our human depth judgment data using linear regression with four degrees of freedom, the affine model allows us to decompose the overall error into distinct, interpretable components: scale ($a_z$), shift ($b$), horizontal shear ($a_x$), vertical shear ($a_y$), and a residual error term. This decomposition is critical as it enables us to disentangle and quantify different sources of systematic bias in human depth perception, moving beyond a monolithic error measure. Importantly, the utility of the affine model is not merely theoretical; it is grounded in a substantial body of several empirical findings across various perceptual tasks [11,50,56,57].

To capture both the non-linear depth compression effect and potential affine transformations in human depth judgments, we adopted an exponential-affine model. Our model incorporates an initial exponential compression stage, estimated across all data points, followed by an affine transformation, fitted independently for each image. Specifically, we first calculated the exponent term $\gamma$ using exponential fitting (Eq (1)) across all data points for each pseudo-participant data. Using this estimated $\gamma$ value, we then performed affine fitting separately for each image to determine affine coefficients: scale $a_z$, shift $b$, horizontal shear $a_x$, vertical shear $a_y$. This fitting was performed using the eight data points ($i = 0, \dots, 7$) within each image for each pseudo-participant data, utilizing the equation:

$$z_{\text{ans}(i)} \simeq a_z * z_{\text{GT}(i)}^{\gamma} + a_x * p_{x(i)} + a_y * p_{y(i)} + b, \tag{2}$$

where $p_{x(i)}$ and $p_{y(i)}$ denote horizontal and vertical image coordinates of each data point $i$, respectively. To avoid negative scaling, which is not perceptually meaningful in this context, we constrained the scale coefficient to be positive: $a_z > 0.0$. Prior to fitting, image coordinates were normalized to $[-1,1]$. Finally, we calculated the residual error for each data point as the difference between the predicted depth and the actual human judgment: $z_{\text{res}(i)} = z_{\text{pred}(i)} - z_{\text{ans}(i)}$, where the predicted depth $z_{\text{pred}(i)}$ was computed using the best-fit parameters $(\gamma^*, a_z^*, b^*, a_x^*, a_y^*)$ as $z_{\text{pred}(i)} = a_z^* * z_{\text{GT}(i)}^{\gamma^*} + a_x^* * p_{x(i)} + a_y^* * p_{y(i)} + b^*$. Fig 1 (the bottom of the column labeled "Decomposing Errors") schematically illustrates how each affine coefficient transforms the ground truth depth.

The results of this fitting analysis are summarized in Table 1. We first observed that the RMS of the residual error after exponential-affine fitting was markedly reduced compared to the original RMSE between human judgments and ground truth depth. This substantial reduction in error indicates that the exponential-affine model effectively captures a significant portion of the systematic biases present in human depth perception data, as we will demonstrate in a complementary analysis in the subsequent section.

Turning to the affine coefficients themselves, we next examine their overall trends by analyzing the mean values across all data points (see the middle column of Table 1). For the scale

**Table 1. Results of affine components and inter-individual similarity of human depth judgments.**

| Measures | Values [95%CI] | Similarity [95%CI] ↑ |
|---|---|---|
| raw error | 1.26, [1.20, 1.33] | 0.808, [0.766, 0.845] |
| $a_z$ | 1.14, [1.01, 1.29] | 0.404, [0.291, 0.514] |
| $b$ | 1.26, [1.07, 1.43] | 0.444, [0.340, 0.545] |
| $a_x$ | −0.0120, [−0.0331, 0.00913] | 0.577, [0.528, 0.627] |
| $a_y$ | −2.03, [−2.11, −1.96] | 0.452, [0.357, 0.540] |
| residual error | 0.191, [0.182, 0.200] | 0.429, [0.401, 0.457] |

and shift components, the average scale coefficient is slightly above unity ($a_z$ = 1.14), while the average shift coefficient is also positive ($b$ = 1.26). This trend suggests that, on average, humans tend to overestimate the depth, particularly in the near distance (note that the model includes the exponential compression factor). While these average scale and shift values differ numerically from the corresponding coefficients derived from the global exponential fitting (scale: $C$ = 2.37 and shift: $\beta$ = −0.808), such numerical differences are understandable given that the affine model, with its additional degrees of freedom including shear, provides a more complex and flexible fit at the image level.

For the shear components, we found a strong bias in vertical shear ($a_y$) compared to horizontal shear ($a_x$). The average value of $a_y$ indicates that, for points at the same ground truth depth, those positioned at the bottom of the image are perceived as significantly closer—by over four meters—than those at the top. This vertical depth bias has been previously identified as a perceptual prior influencing figure-ground segmentation and depth perception [58–60], particularly in studies using artificial stimuli. Our findings extend this understanding by demonstrating that this perceptual bias similarly affects depth judgments in natural indoor images.

To assess how consistently these affine biases varied from image to image across participants, we computed Pearson correlation coefficients for the four affine coefficients between pseudo-participants derived from group averages of random half-splits. Specifically, for each coefficient, we formed a vector containing its estimated coefficient value for each of the 654 images for one pseudo-participant group, and a corresponding vector for the other group. We then computed the Pearson correlation between these two vectors of per-image coefficients. As a result, we found significant positive correlations for all four affine coefficients (the rightmost column of Table 1), indicating that per-image affine components remain consistent across participants when making depth judgments from images. These results align with those of Koenderink et al. [56], who also reported consistent scale, shift, and shear components among participants judging the depth of sculptures from photographs.

Interestingly, although the mean value of $a_x$ revealed no clear systematic bias, its correlation across pseudo-participants at the per-image level was approximately 0.577 for human data, which was higher than that of the other coefficients. This strong positive correlation suggests that horizontal depth errors, such as perceiving one of the left or right points as being more closer than its actual physical position, occur consistently across participants. Although the exact cause remains unclear, it may reflect shared interpretations of image-specific geometric cues or perspective regularities present in each image when inferring the 3D structure.

Finally, to assess the consistency of residual error patterns across participants, we computed Pearson correlations between residual error vectors of pseudo-participants, each derived from group averages of random half-splits. As a result, we observed high correlations in these residual error patterns, suggesting that, even after accounting for the compression effect and affine transformations, systematic deviations persist beyond simple uncertainty.

This suggests that human depth judgments are influenced by additional systematic biases that cannot be fully explained by depth compression and affine transformations, even though these residual errors are small compared to the overall error.

**2.2.4. Affine transformations effectively capture a significant portion of human biases.** Having identified that human depth judgments exhibit systematic biases related to depth compression and affine transformations, we now investigate the extent to which these transformations can account for the variance in these judgments. To that end, we compared the goodness-of-fit of four progressively more complex regression models:

**Exponential fitting** captures global depth compression across the entire dataset:
$$z_{ans} \simeq C * z_{GT}^{\gamma} + \beta$$
**Scale-shift fitting** accounts for per-image scale and shift, but not shear:
$$z_{ans} \simeq a_z * z_{GT} + b$$
**Affine fitting** includes per-image scale, shift, and horizontal/vertical shear, but no global compression:
$$z_{ans} \simeq a_z * z_{GT} + a_x * p_x + a_y * p_y + b$$
**Exponential-affine fitting** combines global compression with affine transformations:
$$z_{ans} \simeq a_z * z_{GT}^{\gamma} + a_x * p_x + a_y * p_y + b$$

To evaluate how much of the total variance in human depth judgments each model captures, we first calculated the coefficient of determination ($R^2$) for each fitting model. The left panel of Fig 3 displays the $R^2$ values for each model. It shows that exponential fitting accounts for only a modest portion of the overall variance (0.469), whereas scale-shift explains a notably larger proportion (0.715). By contrast, both affine and exponential-affine fittings capture a substantial majority of the variance in human depth judgments (0.985), highlighting the critical role of per-image affine transformations.

To further examine how these models align with the consistent bias patterns shared by different observers, we computed Pearson partial correlations while controlling for the ground truth as a covariate. Specifically, for each random half-split of human participants, we replaced one split's depth judgments with the model's estimates derived from that split, while retaining the other split's original data. We then computed the partial correlation between these paired sets. This random half-split approach, analogous to cross-validation, allows us to evaluate the generalizability of the fitting models and assess whether they capture robust

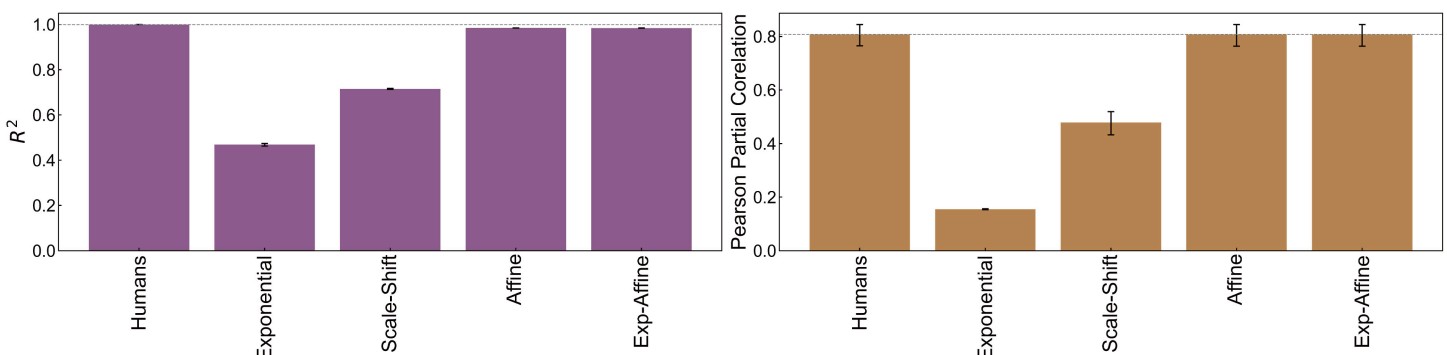

**Fig 3. Performance of regression models in capturing human depth judgments.** The left panel presents the coefficients of determination ($R^2$), and the right panel illustrates the similarity to human depth judgments, measured by Pearson partial correlations (controlling for ground truth) between model estimates and human data. Error bars denote the 95% confidence intervals from 1,000 random half-splits of participants.

error patterns that generalize across different subsets of human observers, rather than simply overfitting to specific individuals or data points. The right panel of Fig 3 shows the results. Exponential fitting showed a low partial correlation (0.155). Scale-shift fitting improved this somewhat (0.479). However, affine fitting and exponential-affine fitting achieved significantly higher partial correlations, both at 0.807. Notably, these values are nearly equivalent to the inter-individual partial correlation observed within human data itself.

In summary, these findings suggest that a model estimating affine parameters for each image can effectively capture the primary biases in human depth judgments. Nonetheless, as noted earlier, some structured errors persist in the residuals (Table 1), indicating that human depth perception involves additional subtleties not fully explained by the affine transformations alone.

## 2.3. How similar are depth judgments between humans and DNNs?

Having established the systematic biases observed in human depth judgments, we now investigate whether DNNs, trained to estimate depth from images, demonstrate similar tendencies. To answer this, we conducted a comprehensive analysis across a diverse set of 64 pretrained models (see Materials and methods, Table 2).

This diverse model selection is crucial for exploring a wide range of performance levels and potential biases. We categorized the models along three key dimensions: learning paradigm, architecture, and dataset. First, we considered the learning paradigm: Supervised models take RGB images as input and are trained directly with ground truth depth maps, while self-supervised models learn without explicit depth labels, by leveraging signals derived from image sequences or stereo images (i.e., disparity and camera pose) during training. At inference time, self-supervised models need only a single image to predict disparity, enabling monocular depth estimation without multiple input frames. In addition, we included hybrid models that combine both supervised and self-supervised paradigms. Among these, 'hybrid (disparity)' models incorporate stereo disparity alongside supervised components, whereas 'hybrid (semantic)' models use semantic feature representations derived from large-scale self-supervised contrastive learning (a technique where the model learns to distinguish similar images from dissimilar ones without explicit labels, e.g., [61]). We also tested a couple of models that employ generative modeling for depth estimation, specifically those based on recent advances in diffusion models [62], which fall under the 'generative' category. Second, we considered the underlying model architecture, including CNN, Transformer (known for capturing long-range dependencies [63]), CNN+Transformer hybrids, and the aforementioned Diffusion-based models. Finally, we considered the datasets on which these models were trained. We prioritized testing models trained on both the indoor NYU dataset and the outdoor KITTI dataset, a widely tested benchmark for outdoor depth estimation [64]. Beyond these core datasets, our evaluation also incorporated models trained on a variety of other datasets, including indoor datasets (Bonn [65] and TUM [66]), an outdoor dataset (DDAD [67]), and models trained on even larger and more diverse datasets. Further details are provided in Materials and methods.

A key challenge in comparing human and DNN depth judgments is the difference in their output formats. Humans provide absolute depth judgments (in meters), while some DNN models generate depth maps in the form of disparity or scale-shift invariant depth, which is defined only up to an unknown scaling factor and offset. To address this, we performed our analysis in two stages. First, we focused on the 31 DNN models that natively output absolute depth estimates. This allowed for a direct initial comparison with human judgments. For the remaining models to enable a comprehensive comparison across all 64 DNNs, we

implemented a scale recovery procedure. We derived metric-aligned pseudo-absolute depth data by linearly regressing each DNN's output depth map (for all 64 models) against the ground truth depth map for each image (see Materials and methods for details). Then, we compared these scale-recovered DNN outputs to raw random half-split human judgments (i.e., pseudo-participants' data), keeping our human baseline in its original, absolute scale.

In the following sections, we use a color-coding scheme to visually represent the training data used for each DNN model. Red represents models trained solely on NYU dataset, while orange denotes models trained on multiple datasets that include NYU. Light blue indicates models trained on a single indoor dataset other than NYU, blue corresponds to models trained on a single outdoor dataset, and green represents models trained on multiple datasets excluding NYU. We also use a shortened version of each model's full name, combining the model name, training strategy, the number of parameters (in millions, denoted by "M"), and suffixes indicating the training datasets. For example, {modelname}-tf-1100M-NK indicates a model named 'modelname', using Transformer-based backbone, with 1,100 million parameters, trained on both NYU Depth V2 (N) and KITTI (K). The dataset abbreviations are as follows: B: Bonn [65], D: DDAD [67], K: KITTI [64], N: NYU [48], T: TUM [66]. Similarly, the backbone architecture abbreviations are as follows: 'cn' for CNN, 'tf' for Transformer, 'hy' for a hybrid backbone combining CNN and Transformer, and 'df' for Diffusion. For self-supervised models, it is important to note that while their initial training may not use explicit depth labels, they are typically fine-tuned on one or more labeled datasets; the dataset suffixes in our naming convention refer to these fine-tuning datasets.

**2.3.1. DNN depth judgments partially share human error patterns.** To begin with, we evaluated the accuracy of depth judgments from both DNNs and humans by comparing their predictions to the ground truth depth. For a fair comparison, we calculated RMSE for DNN models using only the same data points within each image for which we collected human judgments. This initial error analysis focused on the 31 DNN models that directly output absolute depth. As illustrated in the left column of Fig 4A, these DNN models exhibited varying levels of accuracy in comparison to human depth judgments. Specifically, models trained on the NYU Depth V2 dataset, or on datasets including it, generally achieved lower errors, surpassing human performance in almost all cases. Conversely, models trained primarily on outdoor datasets tended to show larger errors than humans, reflecting a domain mismatch when applied to indoor scenes. To further investigate, we broadened our analysis to include all 64 DNN models, encompassing those with scale-shift invariant outputs after scaling to metric depth (the right column of Fig 4A). This expanded analysis revealed similar error tendencies related to their training datasets, mostly consistent with our initial findings. Note that the human RMSE is shown only in the left column, as a direct comparison with the models' scale-recovered RMSE would be misleading; the latter is calculated after optimal scaling to the ground truth, whereas the human data retains its inherent scaling biases.

However, simple accuracy metrics like RMSE can be insufficient, as they often saturate for high-performing models. Similarly, raw correlations with both the ground truth and human judgments are also very high for these models, often saturating and thus obscuring nuanced differences in their error patterns (see S1 Fig). To provide a more sensitive measure of human-likeness, we therefore analyze the similarity of the error patterns themselves by computing the Pearson partial correlation, which controls for the ground truth. This is crucial for determining whether human and DNN judgments align merely due to their shared accuracy, or because they share genuine systematic biases. As shown in Fig 4B, most models exhibited partial correlations in the range of approximately 0.15 to 0.2, with the highest value reaching around 0.4. While notably lower than the inter-human partial correlation of

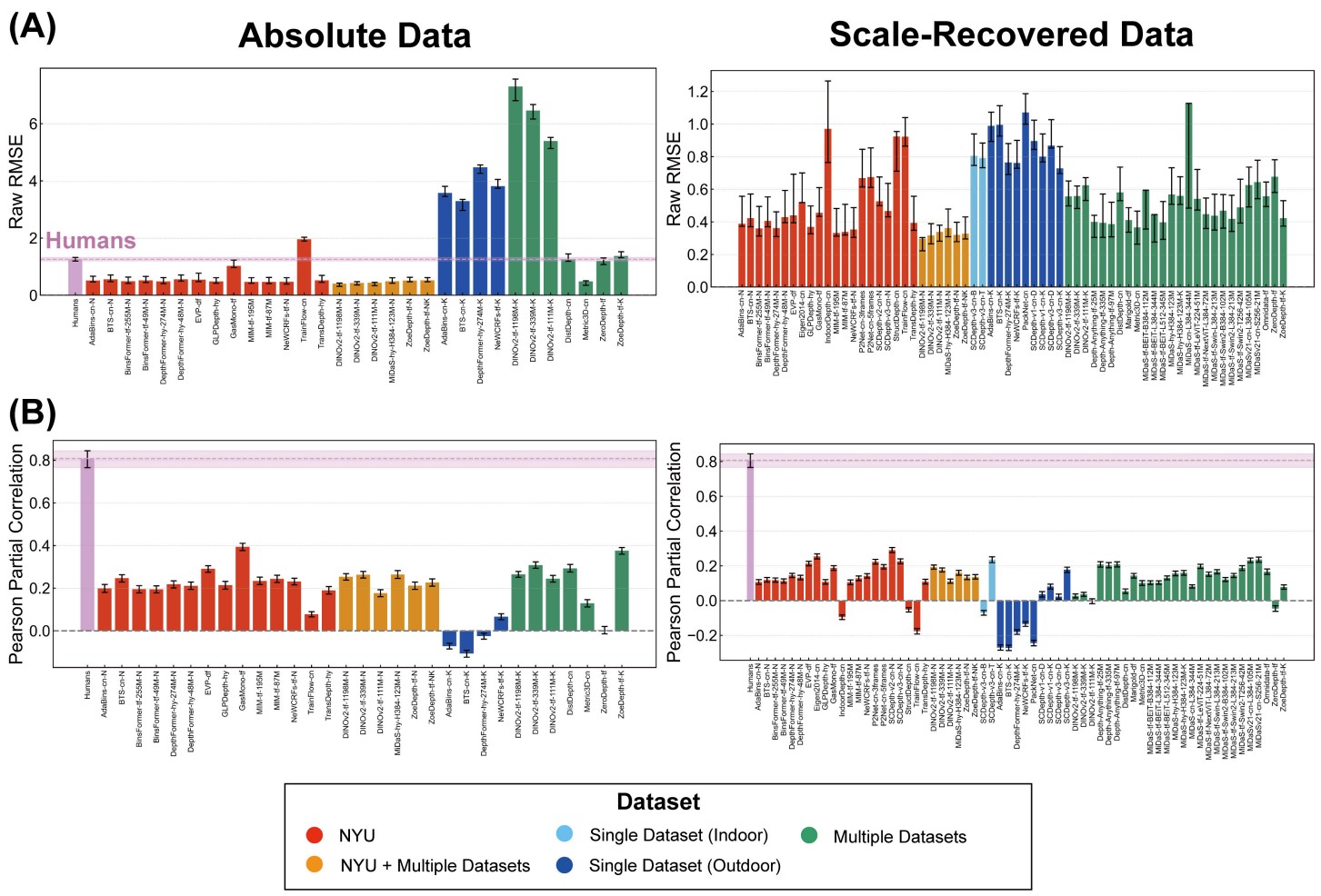

**Fig 4. Comparison of human and DNN error patterns.** (A) Accuracy as measured by RMSE for raw error. (B) Similarity between humans and DNNs based on Pearson partial correlation. For both absolute (left) and scale-recovered (right) analyses, the inter-human partial correlations were calculated from absolute data, serving as a reference benchmark for human-level consistency.

approximately 0.8, these positive values suggest that many DNN models share some systematic error tendencies with human depth judgments. Intriguingly, models trained on single outdoor datasets demonstrated a tendency towards lower partial correlation values compared to human judgments, suggesting qualitatively distinct error patterns.

While the human-DNN comparison reveals a degree of shared bias, it also raises the question of how error patterns vary among DNNs themselves. To investigate this, we computed the Pearson partial correlation between all pairs of 31 absolute-value DNNs and separately among 64 scale-recovered DNNs, then visualized these results as heatmaps (S2 Fig). The heatmap reveals a rich internal structure within the DNN error patterns. Notably, there are visible clusters of high similarity, particularly among models trained on the same i.i.d. datasets (i.e., NYU Depth V2), while models trained only on o.o.d. data (i.e., KITTI) tend to show lower similarity with these clusters. Despite this internal variance, the overall trend is clear. On average, the DNN-DNN similarity is significantly higher than the average human-DNN

similarity in both absolute data (human-DNN similarity: 0.194, 95%CI[0.151,0.238], DNN-DNN similarity: 0.416, 95%CI[0.392,0.440]) and scale-recovered data (human-DNN similarity: 0.097, 95%CI[0.065,0.129], DNN-DNN similarity: 0.435, 95%CI[0.429,0.442]). This indicates that while DNNs do share some error structure with humans, they share substantially more with other DNNs.

To dissect the nature of these shared human-DNN errors more deeply, we analyzed them at a semantic level. We first calculated the per-category RMSE for both humans and each DNN model using the pixel-wise semantic labels from the NYU Depth V2 dataset. Qualitatively, we observed that both humans and DNNs tend to exhibit larger errors for similar categories, particularly those with ambiguous depth cues like reflective surfaces (e.g., mirrors, windows) and ceilings (S3A Fig). This suggests that the shared errors are partly driven by difficulties with categories that are inherently ambiguous or lack strong geometric cues. However, a simple comparison of per-category errors could be confounded by depth: error magnitudes tend to increase with distance, and certain categories naturally appear at different depths. To isolate the effect of semantic category, we quantified this category-level similarity by computing the Pearson partial correlation between the human and DNN per-category RMSEs, while controlling for the average ground truth depth of each category. As shown in S3B Fig, many DNNs showed a positive partial correlation, confirming that they indeed share category-specific vulnerabilities with humans beyond what can be explained by depth alone. Finally, we explored whether this alignment at the category level is linked to a model's overall human-likeness. We compared the model rankings based on this new category-level similarity with the rankings from our main analysis of overall error pattern similarity (point-wise partial correlation). A Spearman rank correlation between these two ranking sets revealed a weak, positive relationship for both absolute data ($\rho = 0.313$, 95%CI [−0.05,0.6]) and scale-recovered data ($\rho = 0.223$, 95%CI [−0.02,0.44]) (S3C Fig). These results suggest a tendency for models with higher overall human similarity to also share the same category-level vulnerabilities, though the modest correlation indicates this link is not definitive and that overall similarity is shaped by more than just shared categorical biases.

**2.3.2. Human-DNN similarity improves with invariant accuracy measures.** To explore the relationship between model accuracy and similarity to human judgments, we examined their correlation using different measures of error. Since it is not a priori clear which error metric best reflects the relevant aspects of depth estimation, we evaluated the relationship across three distinct RMSE spaces: (1) the original metric depth space, (2) the scale-shift invariant space, and (3) the affine invariant space. For each DNN model, we calculated RMSE in these three spaces against the ground truth depth, and correlated these RMSE values with the partial correlation coefficients representing human similarity. For the original metric depth space, we used the RMSE values already computed for the 31 DNNs outputting absolute depth. For the scale-shift invariant space, we used the RMSE of the scale-recovered depth for all 64 DNNs, as described previously. For the affine invariant space, we extended the scale recovery process by fitting not only scale and shift but also horizontal and vertical shear to the ground truth depth for each image, and then computed the RMSE between this affine-transformed DNN output and the ground truth.

The results are presented in Fig 5. The left column displays the relationship between RMSE and human similarity for the 31 DNNs that output absolute metric depth. The right column expands the analysis to all 64 DNN models, including scale-shift invariant models, using scale-recovered depth. In the original metric depth space (Fig 5A), we observed no clear relationship between RMSE and human similarity (Spearman correlation: −0.07, 95% CI [−0.41,0.30]). However, when considering RMSE in the scale-shift invariant space (Fig 5B)

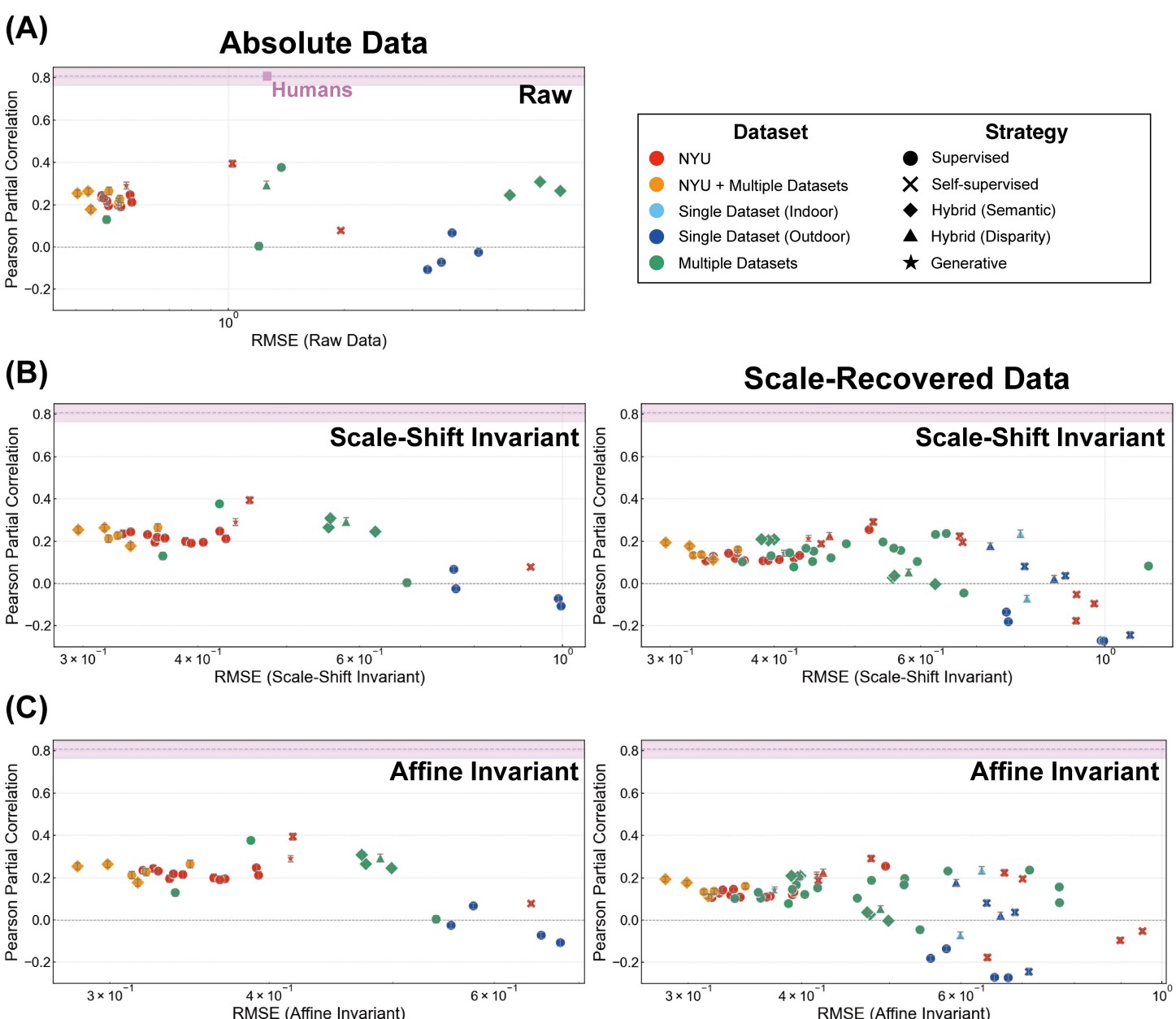

**Fig 5. Scatter plot showing the relationship between RMSE and human similarity.** The horizontal axis denotes (A) the original metric depth RMSE, (B) the scale-shift invariant RMSE, and (C) the affine-invariant RMSE.

and the affine invariant space (Fig 5C), a subtle trend emerged: DNNs with lower RMSE values in these spaces tended to exhibit slightly higher partial correlations with human judgments. For scale-shift invariant RMSE, we found Spearman correlations of –0.28 (95%CI [–0.57,0.09]) for absolute depth and –0.39 (95%CI [–0.58,–0.15]) for scale-recovered depth. For affine-invariant RMSE, we found Spearman correlations of –0.26 (95%CI [–0.56,0.11]) for absolute depth and –0.21 (95%CI [–0.43,0.04]) for scale-recovered depth. This may imply that capturing human-like invariance to scale, shift, and potentially affine transformations is

related to the alignment of DNN error patterns with human judgments. Notably, these conclusions are robust, as we verified that the model similarity rankings remain highly consistent (Spearman's $\rho > 0.96$) even when using the original raw participant data, as opposed to the noise-reduced "pseudo-participant" data generated by our half-split procedure (see S1 Text and S4 Fig).

To evaluate whether the type of depth judgment task influenced our observations, we compared the results from our main absolute depth judgment dataset with those from a supplemental dataset. This supplemental dataset incorporated two additional depth judgment tasks: a two-alternative forced-choice (2AFC) relative depth task and a relative depth magnitude task (see S1 Text for details). Crucially, we found that the relationships between DNNs' RMSE and similarity to human judgments remained largely consistent across these different task types, as illustrated in S5 Fig. This robustness suggests that our findings regarding human-DNN similarity are not an artifact of the absolute depth judgment task itself, but rather reflect more generalizable patterns. It's important to note, however, that the absolute depth judgment task used in our main experiment offers a unique advantage: it allows for both affine decomposition and exponential fitting that is central to our investigation. Relative depth judgment tasks, such as 2AFC, inherently lack absolute depth estimates, making them unsuitable for the kind of quantitative affine analysis that we will show in the following sections.

**2.3.3. Decomposing and comparing error patterns in DNNs.** Following the analysis of overall error similarity, we next sought to understand the nature of DNN error patterns in more detail. To do this, we applied the same exponential-affine model that we used to decompose human depth judgments, allowing us to analyze specific components of systematic bias in DNN depth predictions and compare them to those observed in humans.

*DNNs also exhibit depth compression effect.* To begin, we examined the coefficients of exponential fitting model to investigate how depth compression manifests in the outputs of DNNs. Following the same approach as in the human analysis, we applied exponential fitting to the entire output dataset of each DNN using Eq (1).

Fig 6 presents the exponential fitting coefficients for 31 DNNs that produce absolute depth values. Similar to human depth judgments, most DNNs exhibit an exponent ($\gamma$) less than 1.0 and a scale factor ($C$) greater than 1.0. This pattern suggests a compression of perceived depth at greater distances and an expansion at shorter distances, mirroring the depth compression effect in human vision. However, the magnitude of these biases in DNNs was generally weaker than those observed in human judgments. It should be noted that DNNs, even trained with ground truth depth, independently exhibit depth compression similar to human perception, suggesting potential inductive biases or shared constraints in depth estimation.

We also performed the same analysis on the scale-recovered depth outputs of all 64 DNN models. Here, we focused our analysis on the exponent parameter $\gamma$ of the exponential fitting, as the scale and shift parameters are difficult to interpret due to the influence of the scale recovery process. As shown in S6 Fig, we observed that most DNNs still exhibited $\gamma < 1.0$, similar to the trend seen with absolute depth models. This consistency across model types further highlights the tendency towards depth compression in DNNs.

*Comparison of affine coefficients.* To investigate whether DNNs also exhibit human-like per-image affine biases, we analyzed the affine coefficients (scale: $a_z$, shift: $b$, horizontal shear: $a_x$, and vertical shear: $a_y$) obtained by fitting the exponential-affine model (Eq (2)). Using the same methodology as applied to human data, we first determined the exponent $\gamma$ by the exponential fitting to the entire output of each DNN (Fig 6), and then performed the affine fitting at the eight data points per image where human judgments were collected. The fitting constraints and procedures were identical to those applied to human data.

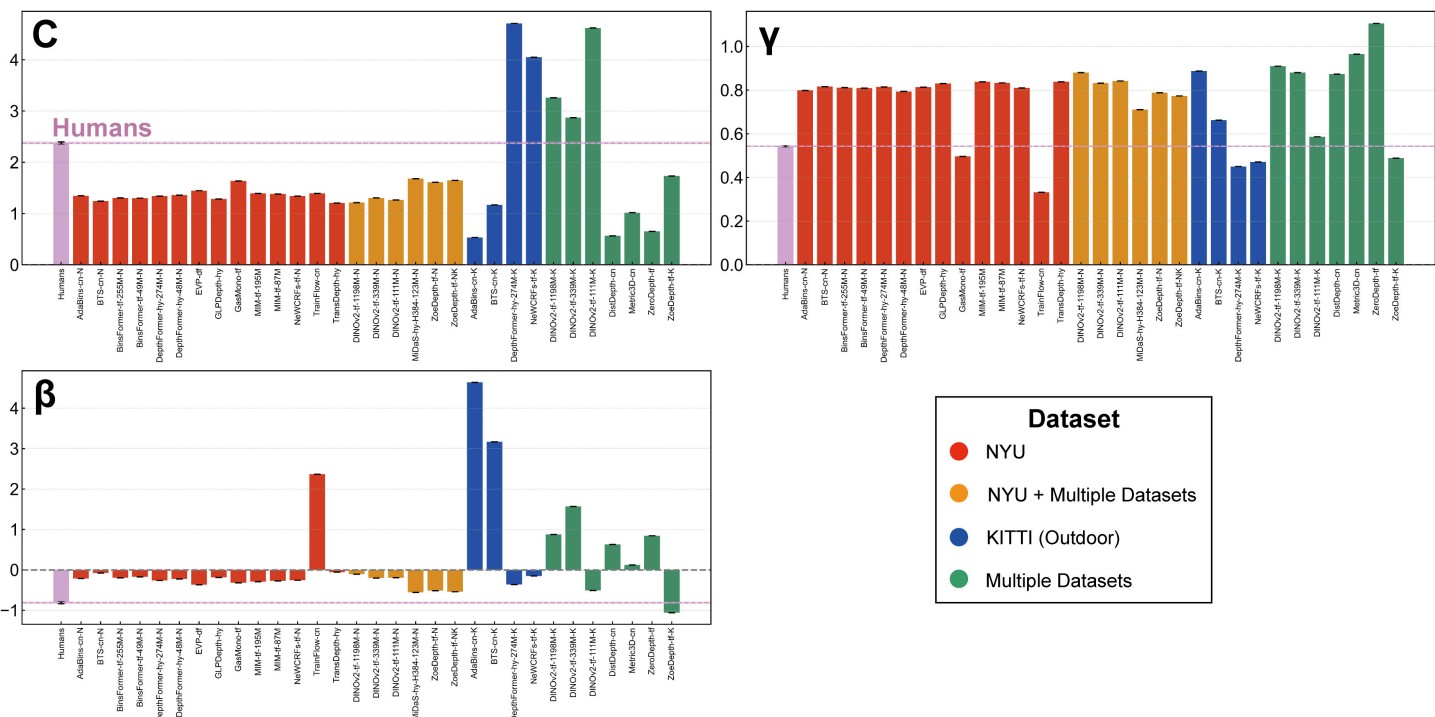

**Fig 6. Exponential fitting coefficients for 31 DNNs that produce absolute depth values.** The figure consists of three subplots: scale component ($C$), exponent component ($\gamma$), and shift component ($\beta$).

Fig 7 presents the four affine coefficients and residual errors for each DNN model that produces absolute depth outputs, averaged across all images. For comparison, the corresponding human values are also shown (pink bar). Examining the scale and shift, we found that most DNNs exhibited $a_z > 1.0$ (scale) and $b < 0.0$ (shift). The value $a_z > 1.0$ suggests that a depth expansion effect at closer distances, resembling the bias observed in human data. Interestingly, the shift component $b$ in DNNs was generally negative, which is a different trend compared to that observed in human depth judgments. However, a notable deviation from this general trend was observed in DNNs trained on the outdoor dataset (blue and green). These models often exhibited substantially larger scale compared to other DNNs and humans, indicating a pronounced depth expansion, which may be attributed to the wider range of depths typically present in outdoor scenes.

For horizontal shear ($a_x$), no significant bias was observed in most DNNs, aligning with human data. In contrast, vertical shear ($a_y$) revealed a notable difference. Humans exhibit a strong negative $a_y$ (lower-region bias), whereas most DNNs, especially those trained on NYU (red and orange) and multiple datasets (green) showed only a slight negative bias, considerably smaller than in human data. This indicates that while DNNs tend to develop a lower-region bias similar to humans, the magnitude is substantially weaker. While most DNNs showed a slight negative vertical shear, some KITTI-trained models (blue) exhibited a positive $a_y$, opposite to the human bias. This reversed vertical shear in KITTI-trained models may also reflect a domain shift, arising from the differences between the outdoor KITTI training data and the indoor NYU test environment.

For residual error, we found that those of DNNs trained on NYU (red and orange) reached a level comparable to that observed for human depth judgments. This similarity in residual

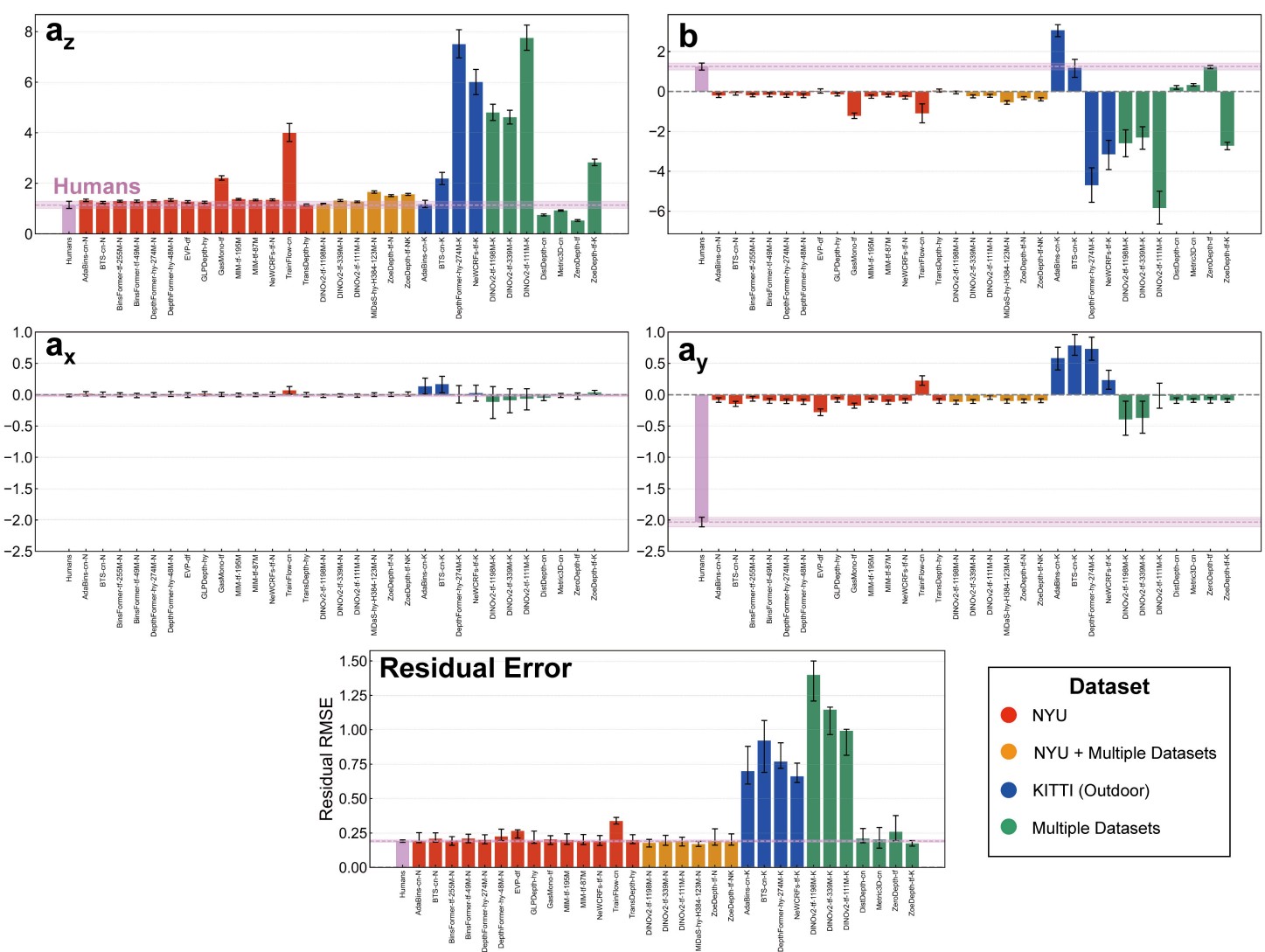

**Fig 7. Averaged affine components for human data and 31 DNNs that produce absolute depth values.** The figure comprises five subplots: scale component ($a_z$), shift component ($b$), horizontal shear component ($a_x$), vertical shear component ($a_y$), and RMSE for residual error. Error bars represent the 95% confidence intervals of the mean values, computed via bootstrap random sampling across individual data points.

error suggests that, within the affine-invariant representation, both humans and these NYU-trained DNNs achieve a similar level of accuracy in capturing the underlying depth structure of the scenes. However, KITTI-trained DNNs (blue) and some DNNs fine-tuned with KITTI (green) exhibited significantly higher residual error. This persistent error in KITTI models suggests they are less effective at capturing the relevant depth structure of NYU test images, even after accounting for affine biases. Their error patterns are not merely affine transformations of ground truth, but reflect a more fundamental mismatch in depth representation, likely due to domain differences in training.

Finally, we extended this affine analysis to the scale-recovered depth outputs of all 64 DNN models (S7 Fig). This broader analysis largely mirrored the trends observed with the metric depth models.

*Per-image affine similarity to humans increases with accuracy.* To assess whether humans and DNNs exhibit similar patterns of affine biases across different images, we compared their per-image affine coefficients and residual error. For each DNN and each coefficient type, we computed the Pearson correlation between two vectors: one vector containing the DNN's coefficient values for all 654 images, and the other containing the corresponding average human coefficient values derived from a random half-split of participants. To ensure robust estimates and 95% confidence intervals, this correlation was calculated 1,000 times using different random half-splits of the human data, and the expected (average) correlations is reported. This analysis quantifies how well the image-by-image fluctuation of each bias component aligns between a DNN and human observers. To explore the relationship between this error pattern similarity and depth estimation accuracy, we then examined these correlation coefficients in relation to the RMSE values of the DNN models, measured in the scale-shift invariant space.

Figs 8A and 9A present the relationship between the scale-shift invariant RMSE of DNNs and their similarity to human judgments for each of the four affine components (scale: $a_z$, shift: $b$, horizontal shear: $a_x$, and vertical shear: $a_y$) and residual error, for both the 31 absolute depths and the 64 scale-recovered depths, respectively. Focusing first on the similarity aspect, we observed modest yet significant positive correlations across all affine coefficients and residual error for most DNNs in both analyses. This indicates that DNNs, regardless of output type, do share systematic bias patterns with human depth judgments for every aspect of affine-transformation related errors as well as the residual error component. Intriguingly, even in scale-recovered data, we found significant positive correlations for scale ($a_z$) and shift ($b$) similarity. This seemingly paradoxical result likely arises because our affine fitting includes shear components. Shared shear biases between humans and DNNs can induce correlated variability in $a_z$ and $b$, even after scale-shift normalization, leading to the observed similarity. Consequently, significant correlation in raw error previously observed between humans and DNNs can now be understood as reflecting these shared affine and residual error characteristics. However, inter-human correlations, represented by the pink line, were consistently and substantially higher than these human-DNN correlations across all measures. This difference underscores that while DNNs exhibit some human-like biases, they only partially capture the full extent and consistency of the systematic biases observed in human depth perception.

Looking at the relationship between DNNs' RMSE and their similarity to human depth judgments, it became apparent that while raw error exhibited relatively low corelation with human similarity (as noted earlier), a clear negative correlation emerged for affine coefficients and residual error. Spearman rank correlation analysis, presented in Figs 8B and 9B, confirmed these negative correlations across all four coefficients and residual error in both analyses. To further assess this consistency, we examined whether DNN models ranked as more or less human-like in one measure also tended to be ranked similarly in others. Indeed, Spearman rank correlations of model rankings across the four affine coefficients and residual error, shown in Figs 8C and 9C, revealed strong positive correlations among these measures. These results strongly suggest that more accurate DNNs (lower RMSE) exhibit systematic biases more aligned with human depth judgments, revealing a link between accuracy and human-like error patterns made clearer by affine decomposition.

*What model features make DNNs more human-like?* To further understand the variations in human-DNN similarity, we investigated potential factors beyond accuracy that might contribute to these differences in error pattern alignment.

To isolate the effect of training datasets, we first compared DNNs with the same architectures but trained on different datasets. Comparing supervised DNNs trained solely on either the NYU dataset (i.i.d. indoor data) or the KITTI dataset (o.o.d. outdoor data) (AdaBins [68],

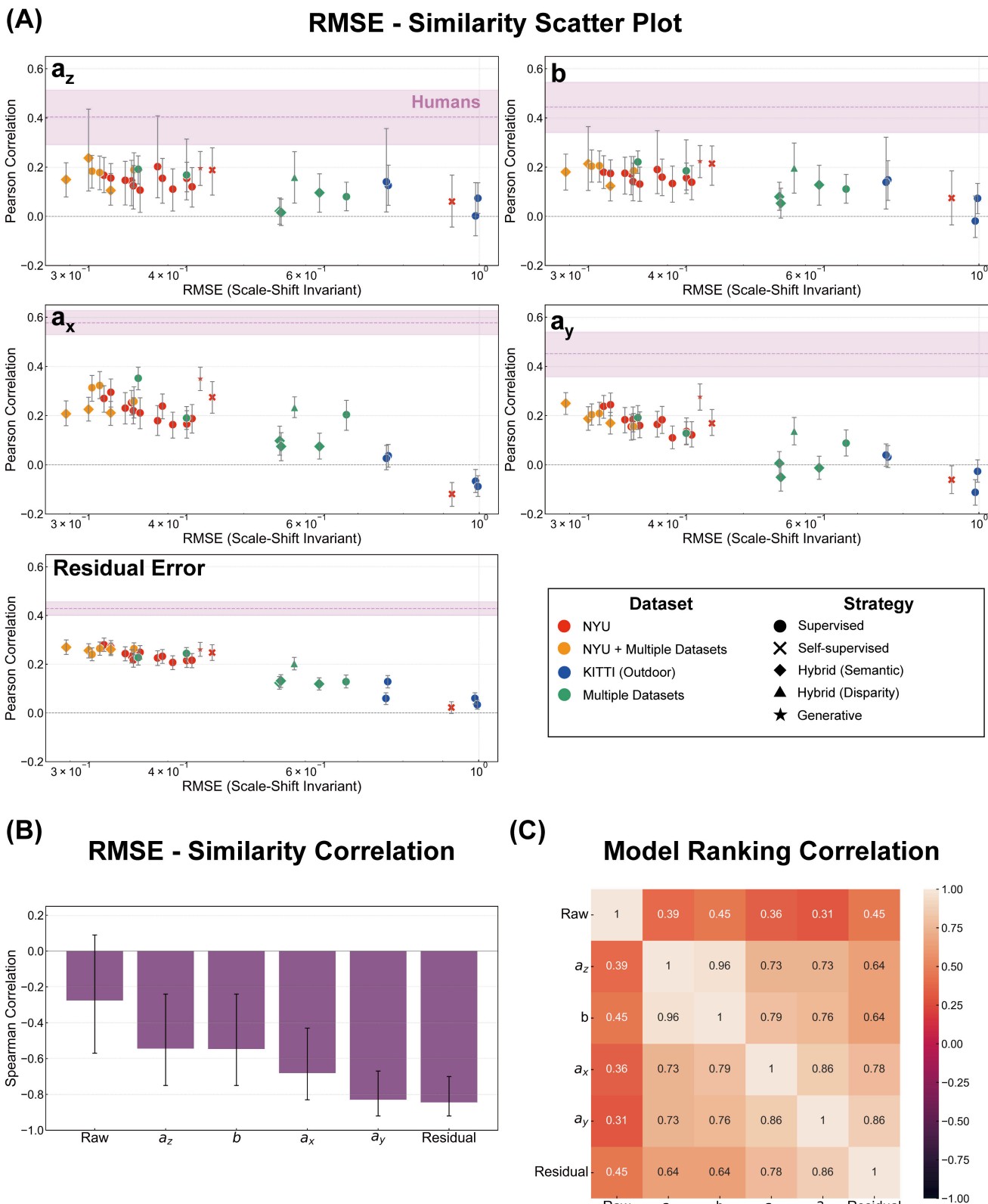

**Fig 8. Relationship between model accuracy and similarity to human judgments for 31 DNNs that output absolute depth values.** (A) Scatter plots illustrating the relationship between scale-shift invariant RMSE and human similarity across different affine components. Marker colors indicate the type of training datasets, while marker shapes represent the training strategy used. (B) Spearman correlation coefficients reflecting the relationship between RMSE and human similarity rankings. (C) Spearman correlation coefficients for model rankings based on human similarity across affine components.

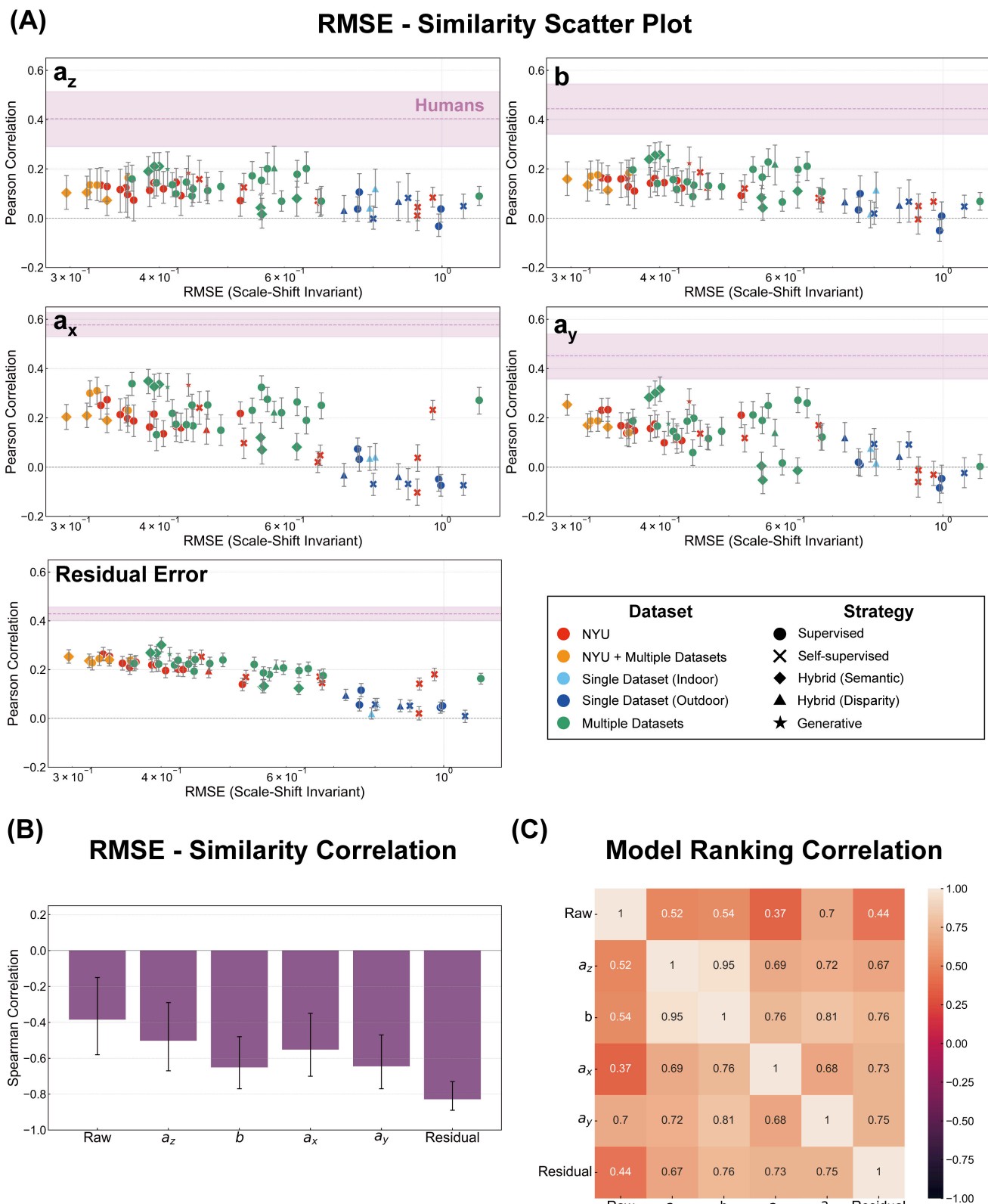

**Fig 9. Relationship between model accuracy and similarity to human judgments for 64 DNNs using scale-recovered data.** (A) Scatter plots illustrating the relationship between scale-shift invariant RMSE and human similarity across different affine components. Marker colors indicate the type of training datasets, while marker shapes represent the training strategy used. (B) Spearman correlation coefficients reflecting the relationship between RMSE and human similarity rankings. (C) Spearman correlation coefficients for model rankings based on human similarity across affine components.

BTS [69], DepthFormer-274M [70], NeWCRFs [71]; Fig 10A), we observed a clear trend: DNNs trained on NYU data consistently exhibited higher similarity scores to human depth judgments than the DNNs trained on KITTI data for both absolute and scale-recovered data. This suggests that DNNs trained on an i.i.d. dataset tend to develop error patterns and coefficients that more closely resemble those of human judgments. Conversely, DNNs trained on KITTI, which consists of in-vehicle images, likely learned a statistical distribution that differs significantly from indoor environments. These results suggest that training solely on datasets with distinct environmental characteristics is insufficient for acquiring depth representations that align with human judgments in a different domain.

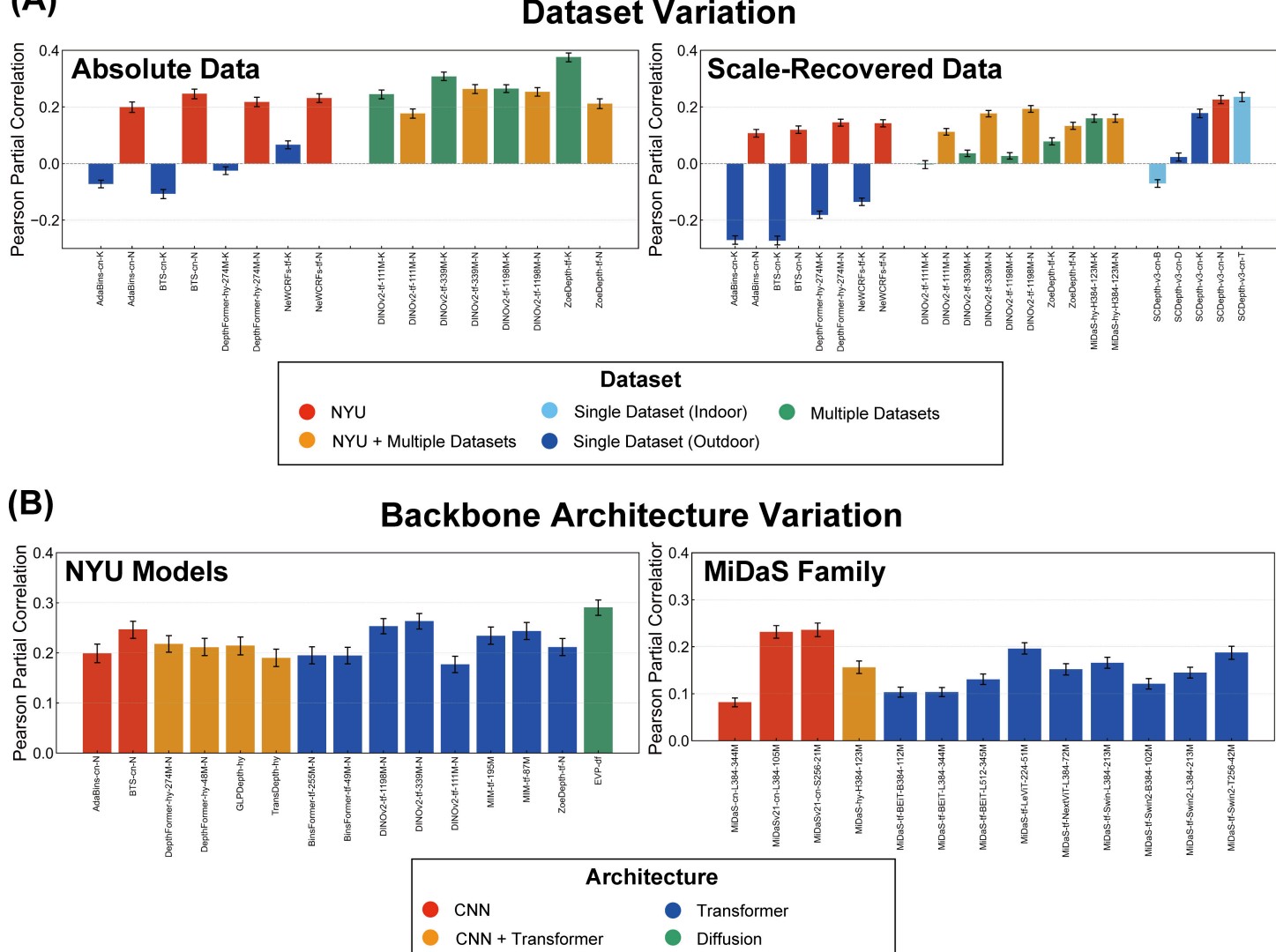

**Fig 10. Human-DNN similarity for selected models.** (A) Variation across different training datasets, with marker colors indicating dataset type. (B) Variation across different backbone architectures, with marker colors indicating architecture type.

DNNs pretrained on multiple datasets and fine-tuned with the NYU or KITTI datasets (DINOv2-111M, DINOv2-339M, DINOv2-1198M [72], ZoeDepth [73], MiDaS-H384-123M [74]) exhibit more complicated tendencies (Fig 10A). For absolute depth data, KITTI fine-tuned models demonstrated comparable or even greater human similarity than NYU fine-tuned models. However, this trend reversed when similarity was measured with scale-recovered data, where NYU fine-tuned models matched or exceeded the human similarity of KITTI fine-tuned models. This suggests that while pre-training on diverse, large-scale datasets enhances general alignment with human judgments, fine-tuning on an i.i.d. dataset like NYU becomes crucial for achieving higher accuracy and scale-shift invariant similarity within that specific domain.

The SCDepth-v3 family of self-supervised models [75] is particularly interesting for assessing the impact of learning strategies, especially considering the biological plausibility of disparity-based self-supervision compared to depth-label supervision. While self-supervised training using disparity information might be hypothesized to promote human-like depth perception, our results for SCDepth-v3 models reveal a more nuanced picture (Figs 9A and 10A). Consistent with supervised models, SCDepth-v3 models fine-tuned on NYU data exhibit relatively higher accuracy and human similarity. Interestingly, KITTI-fine-tuned SCDepth-v3 also achieve a relatively high degree of human similarity, approaching NYU levels, suggesting some human-like error patterns even with o.o.d. fine-tuning. However, SCDepth-v3 models fine-tuned on other indoor datasets (TUM, Bonn) exhibit strikingly low human similarity, failing to align with human judgments in our NYU-based evaluation. This pattern highlights that fine-tuning dataset remains dominant for human alignment, even in self-supervised models. While disparity-based self-supervision in SCDepth-v3 may benefit general depth estimation, it doesn't inherently guarantee or consistently boost human similarity across diverse fine-tuning datasets. Other self-supervised DNNs in our study also reveals a similar lack of consistent human-similarity advantage for self-supervision itself (Fig 9A). While human similarity varies considerably across different self-supervised models, with some achieving relatively high scores (and indeed, the highest similarity in our overall model set is observed in one self-supervised model: GasMono [76]), others show only modest or low similarity, comparable to or even lower than some supervised models.

To assess the impact of model architecture, we focused on two groups of DNNs selected to allow for a somewhat controlled comparison: NYU-fine-tuned DNNs (various architectures fine-tuned on the same NYU dataset) and the MiDaS family (models trained on the same diverse MiDaS dataset mixture, but with different architectures). Examining the human similarity scores within these families (Fig 10B), we observed limited impact of model architecture on the similarity between DNN prediction and human depth judgments. While learning conditions are not perfectly identical across all these models, these findings suggest that, in contrast to training dataset environment and overall model accuracy, the specific choice of high-performing DNN architecture plays a comparatively minor role in determining the alignment of DNN depth predictions with human depth judgments in monocular depth estimation.

**2.3.4. Disentangling metric and ordinal error components by ordinal-level analysis.** Our analyses thus far have focused on the metric properties of depth judgments, revealing shared systematic biases related to depth compression and affine transformations between humans and many DNNs. However, a complete understanding of depth perception requires evaluating not only the accuracy of absolute or relative distance estimates but also the ability to correctly perceive the ordinal relationships between objects (i.e., which object is closer). Do humans and DNNs exhibit similar capabilities and error patterns when judging depth order, potentially independent of the metric biases captured by our exponential-affine model? To

investigate this crucial aspect, we now extend our comparison to the ordinal level. This analysis allows us to directly assess the alignment between humans and DNNs in their judgments of relative depth order, providing insights complementary to the metric-level findings. Furthermore, examining ordinal error patterns, especially within the affine-invariant space (after accounting for affine transformations), allows us to determine whether the observed similarity in metric residual errors between humans and DNNs (Figs 8A and 9A) stems from genuinely shared biases in judging depth order, or simply reflects correlated quantitative inaccuracies that occur even when their perception of relative order might differ.

To compare the similarity of error patterns at the ordinal level, we generated ordinal data from the absolute depth judgments of both humans and DNNs. **Ordinal Data from Absolute Depth (rank)** was generated by converting sets of four absolute depth judgments derived from simultaneously presented data points (human) or corresponding DNN depth predictions, along with ground truth depths, into six pairwise ordinal data. For each pair of points, ordinal data was 1 if the first point was judged closer, and 0 otherwise. To isolate the ordinal error component that could not be explained by the exponential-affine model, we also constructed **Ordinal Data from Residual Errors (residual-rank)**, defined as instances where the ordinal relationships in the raw data from humans and DNNs disagreed with the corresponding predictions of the exponential-affine model. Here, a match between the original judgments and predictions by the exponential-affine model yielded 1, otherwise 0.

After constructing the ordinal data, we computed the error rate, defined as the proportion of incorrect responses in the pairwise data, as a measure of accuracy. To assess the similarity of ordinal error patterns between humans and DNNs, we employed error consistency [30]. This measure quantifies the degree to which errors are shared above chance levels, accounting for overall accuracy. Let $c_{\mathrm{obs}}$ be the observed proportion for shared errors and $c_{\mathrm{exp}}$ the expected proportion of errors occurring at random under the same overall accuracy, error consistency $\kappa$ is calculated as:

$$\kappa = \frac{c_{\mathrm{obs}} - c_{\mathrm{exp}}}{1 - c_{\mathrm{exp}}}. \tag{3}$$

**2.3.5. Humans achieve veridical depth order perception in affine-invariant space.** We first compared ordinal error rates for humans and DNNs in both raw and residual data. The human ordinal error rate in raw depth judgments was 0.254 (95%CI [0.249,0.259]), which significantly decreased to 0.0353 (95%CI [0.0317,0.0391]) for residual data (Fig 11A). Notably, human residual ordinal errors, which represent errors unexplained by the exponential-affine model, were lower than those of all 31 DNNs, even surpassing the accuracy of DNNs trained on the NYU dataset itself. This remarkably low ordinal error rate in the affine-invariant space suggests that human depth perception achieves a highly veridical representation of depth order once affine transformations are accounted for—a finding consistent with the hypothesis that human vision relies on affine-invariant representations of 3D structure [11]. Since only horizontal and vertical shear affect pairwise order, human ordinal errors can be largely attributed to these affine components.

**2.3.6. Shared metric residual error biases do not imply shared ordinal error biases.** To examine the relationship between RMSE in raw data and human similarity across different measures, we employed scatter plots for visualization (Fig 11B). Focusing on the degree of human similarity, we observed that the similarity values from the random half-split of human data, represented by pink line, consistently exceeded the similarity between humans and DNNs, even at the ordinal level. This pattern mirrors the relationship observed in metric-level errors (Fig 8A). Additionally, while metric and ordinal similarity ('raw' and 'rank') exhibited similar RMSE-similarity relationships for raw data, a divergence emerged for residual data.

**(A)**

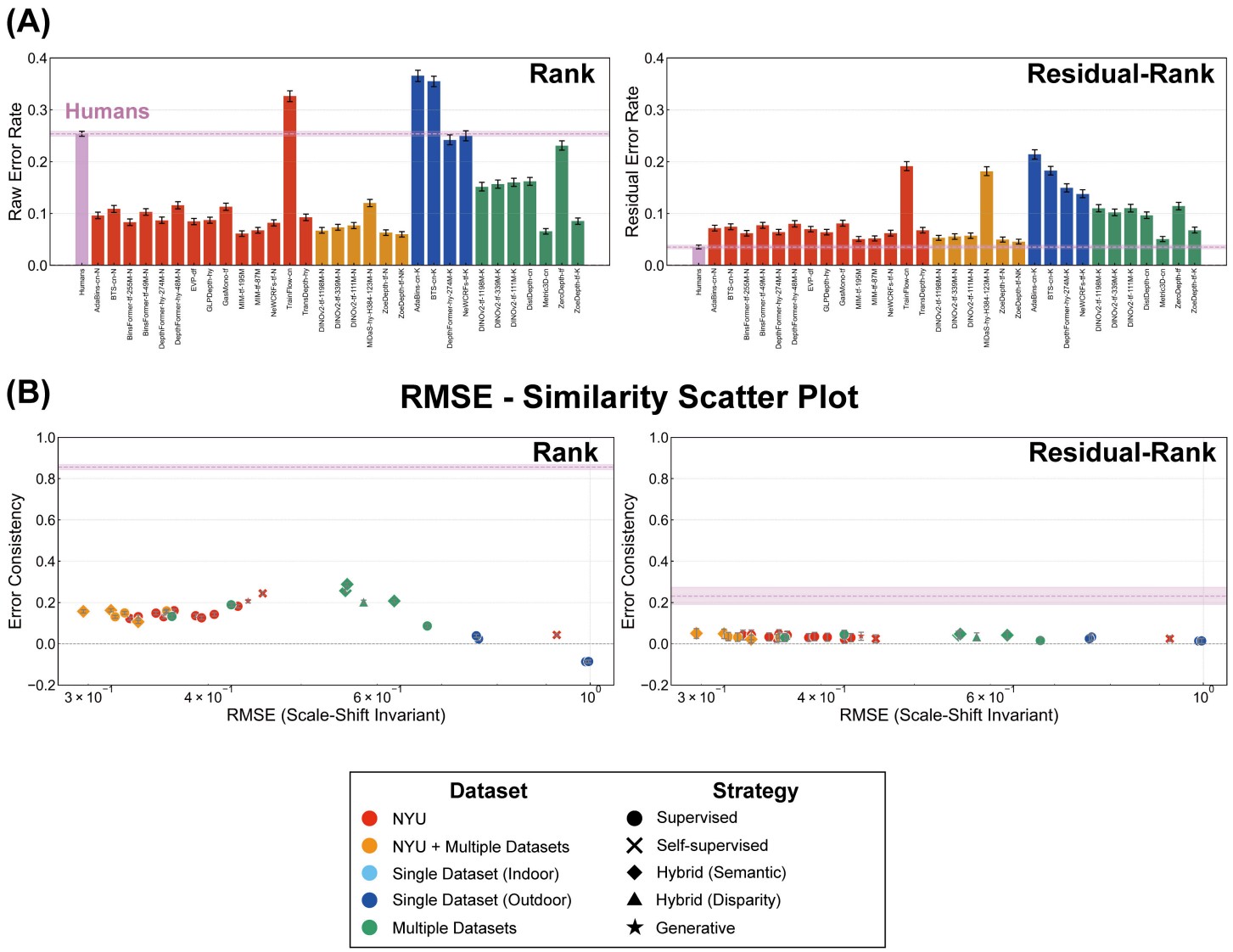

**(B)**

## RMSE - Similarity Scatter Plot

**Fig 11. Results of ordinal-level analyses (rank and residual-rank) for humans and 31 DNNs producing absolute depth values.** (A) Ordinal error rates of humans and DNNs for raw and residual data. (B) Scatter plots illustrating the relationship between scale-shift invariant RMSE and human similarity. Marker colors indicate the type of training dataset used, while marker shapes represent the training strategies employed.

Specifically, for residual data, while metric similarity ('residual') increased with DNN accuracy, ordinal similarity ('residual-rank') remained consistently low across all DNN models, regardless of their accuracy (Fig 11B). This divergence was further confirmed by the Spearman rank correlations between RMSE and the similarity measures ('residual': −0.84, 95%CI [−0.92,−0.70]; 'residual-rank': −0.44, 95%CI [−0.69,−0.10]). This suggests that the positive correlation in metric residual errors between humans and DNNs is not driven by shared ordinal-level biases.

## 3. Discussion

Understanding how both humans and artificial intelligence perceive the 3D world from 2D images is crucial for advancing our understanding of biological vision, with significant implications for the field of computer vision as well. While DNNs have achieved remarkable success in monocular depth estimation, the nature of their depth judgments, especially in comparison to humans, remains underexplored. Previous studies often relied on relative depth judgment tasks, which may not fully capture the nuances of human depth perception, particularly quantitative biases. To address this gap, we constructed a human-annotated dataset of indoor scenes and systematically compared absolute depth judgments between humans and a wide range of DNN models. To investigate the sources of biases in both humans and DNNs, we employed exponential-affine fitting, which captures depth compression effects, linear stretching, and linear shearing. Our results revealed that human depth judgments exhibited systematic biases, including not only depth compression effects and lower-region biases, but also consistent per-image affine distortion patterns. Intriguingly, our comparative analysis demonstrated that high-performing DNNs partially shared these human-like biases, exhibiting similar error patterns in affine transformation parameters and residuals. Crucially, we observed a positive correlation between model accuracy and human similarity: more accurate DNNs tended to exhibit error patterns more closely aligned with human judgments. This suggests that the systematic biases in human depth perception may not be mere imperfections, but rather reflect adaptive strategies or priors to extract meaningful 3D structure from inherently ambiguous 2D images.

However, it is important to note that despite these shared biases, a substantial gap remains. Human-human consistency (partial correlation $\approx 0.8$) still far exceeds the highest human-DNN similarity we observed ($\approx 0.4$). Our analysis provides a crucial insight into this gap: DNNs, regardless of their performance, form a distinct cluster of error patterns, being systematically more similar to each other than to humans (S2 Fig). This suggests a nuanced relationship: while increasing model accuracy may push DNNs towards the human strategy in certain aspects, the models are still largely constrained within a solution space that is fundamentally different from that of human vision. Indeed, our analysis suggests that the improvement in human-DNN similarity with increasing model accuracy may be approaching saturation, particularly within models trained on NYU data, where differences in similarity are minimal across the higher-performing models. Moreover, the error consistency of ordinal judgment patterns between humans and DNNs in the affine invariant space dropped to near chance levels. Furthermore, and perhaps more strikingly, human ordinal-level accuracy remained higher than that of even the best-performing DNNs in this affine invariant space. This divergence highlights a crucial point: while DNNs effectively learn to mimic human-like metric distortions in depth perception, they do not fully capture the human ability to perceive veridical depth order, especially when affine biases are accounted for. Therefore, by further enhancing the ability of DNNs to represent depth order in an affine-invariant manner and improving their ordinal depth perception within this space, we may be able to realize an image computable model that better represent human-like spatial reasoning ability.

Through exponential-affine fitting, we revealed that human depth judgment errors are systematically structured, primarily stemming from depth compression and affine transformations that likely reflect implicit assumptions about viewpoint and scene geometry. Notably, affine transformations alone, encompassing scale, shift, and shear, accounted for a substantial portion of the variance in human depth judgments, suggesting their effectiveness in characterizing human monocular depth perception in natural images. Interestingly, humans exhibited consistent shear, scale, and shift biases across participants—an outcome consistent with a

previous finding on depth ambiguity in pictorial relief perception [56]. Among the affine coefficients, horizontal shear ($a_x$) showed particularly strong inter-individual correlation across images, indicating a left-right bias that cannot be fully explained by simple center-symmetric factors, which posit that points nearer the image center appear closer [40]. This image-specific consistency suggests that human depth perception is not solely driven by generic priors but is also finely tuned to image content when resolving the inherent ambiguities of monocular depth cues.

Why do DNNs partially share some of the systematic depth judgment biases observed in humans? Notably, DNNs exhibit the depth compression effect, a well-documented nonlinear phenomenon in human perception [2–8]. In human vision, such compression is often explained by optimal integration of uncertain sensory cues at long distances, where priors skewed toward nearer distances dominate perception, leading to a systematic underestimation of far-distances[77,78]. DNNs may similarly encounter reduced reliability in texture or other monocular cues for distant regions, prompting conservative (i.e., compressed) depth estimates when visual information is sparse. Alternatively, a comparable mechanism may arise from the scale-invariant loss functions, which incorporate logarithmic operations emphasizing relative rather than absolute errors [79], or from training with disparity as the target signal, in which far-distance errors are inherently diminished because disparity values become small at large depths. In both cases, errors at greater distances are underweighted, increasing estimation uncertainty and pushing the network to rely more heavily on learned priors that favor nearer distances, thereby reinforcing a compression bias at the far end. Beyond depth compression, we also found a positive correlation in per-image affine coefficients between humans and DNNs, indicating that humans and DNNs partially share biases related to linear stretching and shearing. This is particularly intriguing given that DNNs are not explicitly trained to mimic human biases. Instead, the convergence of human and DNN biases likely stems from the shared challenge of inferring three-dimensional scene structure from inherently ambiguous 2D images. Both humans and DNNs must rely on statistical regularities learned from natural images to resolve this ambiguity. The observed similarities in affine biases may therefore reflect the emergence of similar, statistically grounded priors for depth estimation, whether implicitly learned by DNNs from large datasets or developed through evolution and experience in the human visual system. A particularly exciting future direction lies in uncovering the origins of the image-specific affine biases we observed. A computational search through scene parameters (e.g., geometry, textures, and camera properties) could identify the specific visual configurations that elicit affine errors in DNNs. Subsequently testing humans with these machine-generated "illusions" would provide a deep and causal test of human-AI alignment, revealing whether the human visual system shares the same vulnerabilities.

Our analysis also revealed a noteworthy positive correlation between the accuracy of DNN depth estimation, as measured by scale-shift invariant RMSE, and their similarity to human depth judgments, particularly in terms of affine coefficients and residual error. This finding appears to contrast with observations from object recognition tasks, where studies have reported a trade-off: as ImageNet accuracy surpasses a certain threshold, similarity to human error patterns or perceptual judgments can decrease [26,27,30,80]. However, recent work in object recognition suggests that this trade-off may be specific to in-distribution accuracy. When considering out-of-distribution (o.o.d.) accuracy, for example, on noisy or perturbed images, a positive correlation emerges: models robust to distributional shift, exhibiting higher o.o.d. accuracy, also demonstrate greater alignment with human error patterns [31]. While direct investigations into human similarity in depth estimation are still limited, our findings align with emerging evidence suggesting a link between robustness and human-like inductive

biases in DNNs [81]. As a relevant example in depth estimation, Bae and colleagues demonstrated that in outdoor monocular depth estimation, DNNs robust to environmental variations, such as texture shifts and adverse weather conditions, exhibited a stronger shape-bias in their feature representations [34]. Although their study focused on outdoor scenes and o.o.d. robustness rather than human similarity or i.i.d. accuracy, it hints at a broader principle: DNNs that generalize well may develop more human-like representations. Given the complexity of depth estimation, achieving high accuracy might necessitate learning robust and generalizable depth representations, potentially leading to greater alignment with human perception. The affine biases we observed could reflect such shared strategies. Interestingly, despite the overall positive trend observed when considering affine components, a closer look at the relationship between raw error similarity (before affine decomposition) and invariant accuracy (e.g., raw error similarity vs. scale-shift invariant RMSE; Figs 5B and 11B (rank)) reveals a subtle inverted-U shape. This observation might suggest that, even in depth estimation, an accuracy-similarity trade-off similar to that reported in object recognition could potentially exist, where pushing for maximal in-distribution accuracy might not always lead to increased human-like error patterns, especially before accounting for systematic distortions like affine transformations.

## 3.1. Limitation

We acknowledge that our depth judgment task, while ecologically relevant, deviates from natural egocentric distance estimation, as participants judged depth from images on a flat screen. A key aspect of this deviation is that participants viewed the images binocularly, contrasting with the monocular input to the DNNs. While binocular disparity can indeed act as a flatness cue under sparse conditions [82], its role is not merely a source of cue conflict when viewing cue-rich pictures. Instead, the robustness of human vision in this context is thought to leverage this signal, as binocular disparity is used to estimate the picture's physical slant and actively compensate for geometric distortions [83]. This compensatory mechanism, supported by powerful pictorial cues such as familiar size, allows for a stable interpretation of the depicted scene. Thus, our findings likely capture fundamental aspects of this robust human system and its ability to effectively generalize depth perception from 2D representations.

Another limitation stems from estimating per-image affine coefficients using only eight data points, a constraint imposed by the human data collection feasibility. Fitting a model with four free parameters to such sparse data per image carries a risk of overfitting. Consequently, the explanatory power attributed to simple affine transformations in our analysis might be overestimated for some images, potentially masking more complex or non-homogeneous distortions inherent in the data. While the consistent overall trends revealed across models in our study suggest meaningful structure was captured by this approach, interpreting the precise per-image coefficients requires caution due to this potential limitation. More robust coefficient estimation would require denser human depth judgments per image, which remains a challenge for future work.

Although we used affine transformations as an analytical framework for their simplicity and empirical grounding, other geometric models, such as perspective transformations [6], could also represent visual space. Moreover, the assumption of a globally uniform visual space itself is debatable [1,13,50]. Emerging theories suggest that visual space might be constructed dynamically, with local representations formed at each gaze point and subsequently integrated [10]. Despite accounting for a significant portion of the variance, the affine models leave residual errors, indicating they may not fully encompass the complexity of visual space representation. These residual errors in our study could potentially be explained by these

alternative geometric models or dynamic visual space theories. Future research should explore these more complex representations to refine our understanding of human depth perception and DNN error patterns.

Finally, we note key limitations in the scope of both the data and models analyzed, which point to important avenues for future work. Our study was based exclusively on indoor scenes from the NYU Depth V2 dataset. While this dataset is valuable for controlled comparisons, future studies should investigate a broader range of datasets, including outdoor scenes, to assess the generalizability of our findings and conclusions across diverse visual environments. Likewise, our investigation was largely restricted to discriminative models, aside from a few diffusion-based approaches. These generative approaches, along with others like generalizable NeRFs [84,85], learn powerful priors to generate, not just estimate, plausible 3D structures. Whether this ability to synthesize scenes leads to a better alignment with human perceptual biases, compared to direct regression models, remains a critical question for future work.

## 4. Conclusions

Comparing human and DNN depth judgments is crucial for both vision science and AI development. Our study reveals that while DNNs achieve high accuracy in monocular depth estimation, they also partially replicate human systematic biases, particularly depth compression and some affine distortions. The positive correlation between DNN accuracy and similarity to human error patterns suggests a degree of convergence towards human-like perceptual strategies in better models. However, significant divergences persist. DNNs do not fully match the consistency of human affine biases, and their ordinal depth perception in affine-invariant space remains less human-like. This underscores that current DNNs, despite mimicking some metric distortions, do not fully capture human spatial reasoning. Evaluating AI models beyond veridical accuracy, towards ecological adaptability and human-alignment, is therefore essential. Future development should focus on DNNs better representing affine-invariant depth and improving ordinal perception in this space. Data-driven models explicitly incorporating human perceptual characteristics can lead to more interpretable, reliable, and human-compatible depth perception, benefiting both scientific understanding of vision and real-world AI applications. Further comparative research with broader stimuli and neuroscientific data will be key to realizing this potential.

## 5. Materials and methods

### 5.1. Ethical statement

All psychophysical experiments were conducted in accordance with the Declaration of Helsinki, and their experimental designs were approved by the Institutional Review Board of Communication Science Laboratories, NTT, Inc. For the online experiments, the participants provided their electronic informed consent to participate in this study. Only those who fully understood the content and purpose of the experiment and provided their consent online participated in the experiment.

### 5.2. Stimuli

We selected image stimuli from the NYU Depth V2 dataset, one of the most widely used indoor RGB-D datasets collected using Microsoft Kinect [48]. While newer, higher-resolution datasets, both real and synthetic, have become available (e.g. [86–88]), we opted for the NYU dataset and used 654 test images for our evaluation. This dataset remains a de facto standard benchmark for indoor monocular depth estimation due to its broad representation of diverse

indoor environments. Compared to outdoor, in-vehicle datasets like KITTI [64], the NYU dataset exhibits greater diversity in indoor scene structures, encompassing walls, floors, furniture, and various objects commonly found in indoor spaces. This diversity is crucial for evaluating depth perception in complex, realistic indoor settings, making it well-suited for our human-DNN comparison study.

In this study, we utilized two datasets, both derived from the same set of 654 test images from the NYU Depth V2 dataset. The **main dataset** focused on collecting absolute depth judgments for pre-selected points within each image. This dataset served as the foundation for our primary analyses comparing human and DNN depth perception and investigating systematic biases in depth estimation. To validate the findings from the main dataset and assess the influence of task type, we also collected a **supplemental dataset** focused on relative depth judgments. Previous studies investigating human depth perception have frequently employed relative depth judgment tasks, such as two-alternative forced-choice (2AFC) [39,40], ordinal data collection [42], and relative depth value assessments [53]. To ensure comparability with these established methodologies and assess the potential influence of task type on our findings, we adopted a similar approach for collecting the supplemental dataset. Both datasets thus share the same image stimuli but differ in the type of human depth judgments collected. See S1 Text for the results of the supplemental dataset.

### 5.3. Sampling image locations for depth estimation

**5.3.1. Main dataset (Absolute depth judgments).** For each of the 654 test images, we selected two sets of four points (eight points per image in total) based on the following three criteria: (A) Within each set of four points, each point was separated by at least 20 px both vertically and horizontally to prevent visual interference. (B) Points were positioned at least 20 px away from the image borders to avoid potential edge-related visual artifacts. (C) Points were located at least 5 px away from object segmentation boundaries, derived from the NYU Depth V2 dataset's annotation data, to minimize ambiguity in point-object assignment (border ownership) near depth discontinuities at object edges. To select candidate points efficiently while satisfying these criteria, we implemented the following procedure: when a point $p_i$ was randomly selected within an image, it was evaluated against the previously selected points $[p_0, \ldots, p_{i-1}]$ in the candidate set to ensure all three criteria were met. If all conditions were satisfied, the point was added to the set. If not, a neighboring point within the same segmentation region was searched for as $p_i$ and added if it met the criteria. This intra-region search strategy helped reduce the over-representation of large, homogeneous regions like floors and walls in the selected points. If no suitable point was found within the current segmentation region, the process resumed by sampling new random points until a valid candidate was identified. If the procedure failed to generate a complete set of candidate points, the entire set was discarded, and the selection process was restarted from scratch. Using this procedure, we successfully identified eight candidate points for each of the 654 test images, which were then used for collecting absolute depth judgments from human participants.

**5.3.2. Supplemental dataset (Relative depth judgments).** For the supplemental dataset, we selected four points per image from the same set of 654 test images to collect relative depth judgments. Point selection for the supplemental dataset followed a slightly different approach in terms of criterion (C). For criteria (A) and (B) (spatial separation and distance from image borders), we applied the same pre-processing procedures as in the main dataset. However, for criterion (C) (distance from segmentation boundaries), we did not apply the same pre-processing exclusion as in the main dataset. Instead, we addressed the potential influence of segmentation boundaries during post-processing of the collected human data. As discussed in

the Results section, differences in extraction methods and experiment paradigms between the two datasets did not affect the overall pattern of results or our main conclusions.

## 5.4. Procedures of collecting human dataset

Participants were recruited through a crowd-sourced online survey, with individuals ranging in age from their 20s to 40s (main dataset: 898 participants, supplemental dataset: 536 participants). All observations were conducted binocularly. Prior to the experiment, participants were instructed to disable any night mode or dark mode settings in their browsers and to view the display in full-screen mode. To ensure consistency in stimulus size across participants, they were asked to maintain an observation distance of 60 cm and to adjust a rectangle displayed on the screen until it matched the size of a credit card held in their hand. This process allowed for the calibration of the stimulus size based on the screen's pixel density. As a result, the image was presented at 9.6 cm × 7.2 cm, corresponding to a visual angle of 9.15° × 6.87°. Our crowd-sourcing methodology precluded the collection of detailed participant information (e.g., visual acuity) and specifics of their viewing environment (e.g., precise viewing distance). While these uncontrolled factors could add noise to the data, their influence on our main findings is likely minimal, as our analyses are based on judgments averaged across a large participant pool.

Prior to the main trials, participants underwent example and practice trials (main dataset: five trials, supplemental dataset: seven trials) with feedback to ensure they understood both the task requirements and the response methods. Feedback consisted of presenting ground truth-based example responses below the participants' responses. During the task, participants were instructed to judge the distance of each target point from their viewpoint, imagining themselves as the camera position in the scene. They were asked to provide either absolute (main dataset) or relative (supplemental dataset) distance judgments based on this egocentric perspective. For target points located on reflective or transparent surfaces such as mirrors and glass, participants were instructed to report the distance to the surface itself, consistent with phenomenological distance in daily life. We evaluated the human depth judgments based on depth values after applying geometric correction and missing value interpolation to the raw data collected by Kinect, which is publicly available in the original repository [48]. Stimuli were presented in random order for each participant to minimize order effects. Following data collection, we implemented a data screening procedure to exclude data from participants exhibiting low reliability. The specific experimental procedures and reliability assessment methods for each dataset are detailed below.

**5.4.1. Main dataset (Absolute depth judgments).** Participants were presented with stimuli displaying four target points, labeled A to D. For each point, they reported the perceived distance from their viewpoint (camera position) in meters using an interactive scale bar (see Fig 1 for task interface). Each participant completed 28 or 30 trials of this task. The pre-selected target points, as described in the previous section, were randomly assigned to 46 stimulus sets, ensuring that the two sets of four points from the same image were included in the same stimulus set. Each stimulus set was evaluated by at least 20 participants. To ensure data quality, we excluded data from participants who demonstrated low reliability in their responses. Reliability was quantified using Spearman rank correlation between each participant's responses and the median responses across all participants. When identifying the outlier, we should consider the proportion of participants with low reliability may vary across images. To account for this, we first calculated the Spearman correlation between the median judgments and the individual judgments of each participant within the group that responded to the same stimulus set. Then, we aggregated these correlation values across all participants

and computed the inter-quartile range to assess overall reliability. Participants whose correlation values fell below 1.5 times the inter-quartile range were identified as unreliable and excluded from further analysis. This procedure resulted in a final sample of 742 valid participants out of the initial 898 recruited for the absolute depth judgment task.

**5.4.2. Supplemental dataset (Relative depth judgments).** Participants were presented with stimuli displaying two target points, labeled A and B. For each stimulus, they first responded which point appeared closer to the viewpoint by selecting a radio button. They were instructed to choose one of the two points even if both appeared to be at nearly the same distance, following a two-alternative forced-choice (2AFC) protocol. Subsequently, participants reported the perceived relative distance ratio between the two points by adjusting the length of a movable blue bar relative to a fixed red bar, where the red bar represented the distance to the farther point (see Fig 12 for task interface). Each participant completed 84 or 88 trials of this task. The 2,616 point pairs were randomly divided into 30 stimulus sets, ensuring that all four pairs derived from the same image were included within the same stimulus set. Each stimulus set was evaluated by at least 15 participants. Participants with low reliability in either the 2AFC or proportion data were excluded from subsequent analyses. For the 2AFC data, participants were considered unreliable if their response log-likelihood was more than 1.5 times below the inter-quartile range of the distribution of log-likelihood across all participants. Specifically, for the 2AFC judgments obtained from each participant, we calculated the average selection ratio $p_{ave}^{(i)}$ for each pair of points within each stimulus set. Then, we computed the log-likelihood of each participant's binary choice $p^{(i)} \in \{0, 1\}$ given the group average selection ratio using the formula: $\sum_i p^{(i)} \log(p_{ave}^{(i)} + \epsilon) + (1 - p^{(i)}) \log(1 - p_{ave}^{(i)} + \epsilon)$, where $\epsilon = 0.01$ is a small constant added to ensure numerical stability in the logarithmic calculation. Participants were identified as outliers if their log-likelihood value fell more than 1.5 times below the inter-quartile range. For the proportion data, participants were deemed unreliable if Spearman rank correlation between each participant's responses and the median response fell below 1.5 times the inter-quartile range. Applying these criteria, we retained data from 463 valid participants out of the initial 536 recruited for the relative judgment task.

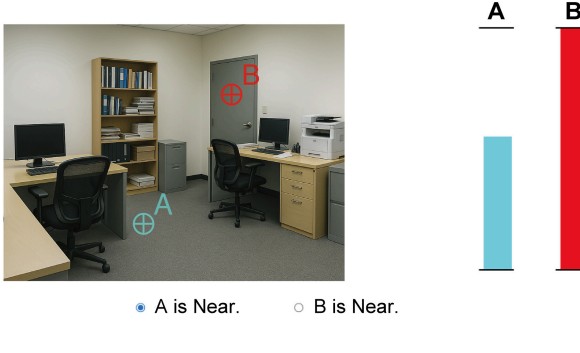

**For Point A and Point B, which point in the screen appears closer to you?**

How close is it to the nearer point compared to the farther point? Please answer it with the scale bar.

○ A is Near.    ○ B is Near.

[ Next ]

**Fig 12. Example of the task screen for collecting supplemental dataset (relative depth judgments).** All instructions were displayed in Japanese in the actual experiment. The image was generated by ChatGPT (GPT-4o, 2025) to avoid potential copyright issues and is intended for illustrative purposes only.

## 5.5. Target DNN models

Table 2 provides a summary of the 64 DNN models evaluated in this study. We recruited a diverse set of 64 pretrained monocular depth estimation models, encompassing a variety of training strategies (supervised learning, self-supervised learning, and generative), backbone architectures (CNN-based, Transformer-based, hybrid (mixture of CNN and Transformers) and Diffusion-based models), training datasets (with or without NYU and KITTI training, as well as training on other datasets), and parameter size. Specifically, training strategies can be broadly categorized into supervised learning, where depth images serve as explicit supervision, and self-supervised learning, which relies on disparity and camera pose estimation instead of direct depth supervision. Beyond these conventional approaches, we included hybrid models that combine aspects of both supervised and self-supervised learning. Within these hybrid approaches, we distinguish two subcategories. 'Hybrid (disparity)' models leverage a combination of supervised training, often using a teacher model trained with depth labels, and self-supervised training components that incorporate stereo disparity estimation. These models, such as DistDepth [89] and SCDepth-v3 [75], utilize stereo pairs to learn depth representations by minimizing disparity-based losses, while also potentially benefiting from knowledge transfer from a depth-supervised teacher network. In contrast, 'hybrid (semantic)' models employ a different form of hybrid learning, leveraging semantic feature representations learned through self-supervised contrastive learning on massive unlabeled image datasets. Models like Depth-Anything [90] and DINOv2 [72] fall into this category. Additionally, we employed a couple of models that integrate generative modeling into the depth estimation task (generative). These generative models, specifically EVP [91] and Marigold [92], both leverage the power of diffusion models, employing Stable Diffusion [93] as a foundational component with a supervised decoder attached to its back end for depth prediction.

Focusing on training datasets, our evaluation includes not only DNN models trained on NYU or KITTI but also those trained on a more diverse range of datasets to allow for a more comprehensive evaluation of DNN characteristics. For models trained on datasets beyond NYU and KITTI, we include the SCDepth family, which incorporates models trained on Bonn [65], TUM [66], and DDAD [67] datasets. The Bonn RGB-D dynamic dataset captures dynamic interactions with moving objects and people in indoor scenes. The TUM RGB-D dataset consists of continuously recorded indoor sequences from a moving camera. The DDAD dataset focuses on long-range depth estimation in urban driving scenarios.

Shifting focus to models trained on multiple datasets, DistDepth [89] is trained on SimSIN, consisting of multiple datasets simulating various indoor scenes (Replica, MP3D, and HM3D). Omnidata [94] primarily learns from synthetic datasets with multi-view images captured from indoor scenes, also incorporating 3D scans of real objects. Moving to models trained on both indoor and outdoor environments, DNN models belonging to MiDaS family [74,95,96] were trained on a foundational dataset (MIX6 or MIX10), encompassing a diverse range of data, including indoor images, multi-view images, web images, 3D movie frames, and stereo images. Leveraging MiDaS dataset, ZoeDepth [73] undergoes pretraining on MIX10 in addition to NYU and KITTI before being fine-tuned on NYU or KITTI, while Depth-Anything [90] is trained on six labeled datasets, which form part of MiDaS dataset, followed by eight unlabeled datasets assigned pseudo-depth label datasets from the teacher model. Meanwhile, Metric3D [97] and ZeroDepth [73] employ different combinations: Metric3D integrates 11 datasets covering both real and CG-generated images across indoor and outdoor scenes, whereas ZeroDepth learns from a mixture of CG-generated driving

**Table 2. Target 64 DNN model list.** The model outputs were categorized based on the type of depth information provided: absolute depth (no symbol), relative depth (indicated by [†]), and disparity (indicated by [‡]). Full version of this table is available in the Supporting Information.

| Models | Strategy | Backbone | Dataset | | | Size |
|---|---|---|---|---|---|---|
| | | | NYU | KITTI | Others | |
| AdaBins-cn-K [68] | supervised | CNN | - | ✓ | - | 78,257,238 |
| AdaBins-cn-N [68] | supervised | CNN | ✓ | - | - | 78,257,238 |
| BinsFormer-tf-49M-N [101] | supervised | Transformer | ✓ | - | - | 49,231,191 |
| BinsFormer-tf-255M-N [101] | supervised | Transformer | ✓ | - | - | 254,625,329 |
| BTS-cn-K [69] | supervised | CNN | - | ✓ | - | 47,481,969 |
| BTS-cn-N [69] | supervised | CNN | ✓ | - | - | 47,481,969 |
| Depth-Anything-tf-25M [90][†] | hybrid (semantic) | Transformer | - | - | 14 datasets | 24,785,089 |
| Depth-Anything-tf-97M [90][†] | hybrid (semantic) | Transformer | - | - | 14 datasets | 97,470,785 |
| Depth-Anything-tf-335M [90][†] | hybrid (semantic) | Transformer | - | - | 14 datasets | 335,315,649 |
| DepthFormer-hy-48M-N [70] | supervised | hybrid | ✓ | - | - | 47,612,029 |
| DepthFormer-hy-274M-K [70] | supervised | hybrid | - | ✓ | - | 273,732,631 |
| DepthFormer-hy-274M-N [70] | supervised | hybrid | ✓ | - | - | 273,732,631 |
| DINOv2-tf-111M-K [72] | hybrid (semantic) | Transformer | - | ✓ | 1.2B data | 110,774,401 |
| DINOv2-tf-111M-N [72] | hybrid (semantic) | Transformer | ✓ | - | 1.2B data | 110,774,401 |
| DINOv2-tf-339M-K [72] | hybrid (semantic) | Transformer | - | ✓ | 1.2B data | 338,558,657 |
| DINOv2-tf-339M-N [72] | hybrid (semantic) | Transformer | ✓ | - | 1.2B data | 338,558,657 |
| DINOv2-tf-1198M-K [72] | hybrid (semantic) | Transformer | - | ✓ | 1.2B data | 1,198,281,537 |
| DINOv2-tf-1198M-K [72] | hybrid (semantic) | Transformer | ✓ | - | 1.2B data | 1,198,281,537 |
| DistDepth-cn [89] | hybrid (disparity) | CNN | - | ✓ | 3 datasets | 69,206,908 |
| EVP-df [91] | generative | Diffusion | ✓ | - | - | 933,814,544 |
| Eigen2014-cn [79][†] | supervised | CNN | ✓ | - | - | 240,833,218 |
| GLPDepth-hy [102] | supervised | hybrid | ✓ | - | - | 61,220,903 |
| GasMono-tf [76] | self-supervised | Transformer | ✓ | - | - | 28,000,697 |
| IndoorDepth-cn [103][†] | self-supervised | CNN | ✓ | - | - | 14,840,507 |
| Marigold-df [92][†] | generative | Diffusion | - | - | Hypersim, Virtual KITTI | 1,289,963,947 |
| Metric3D-cn [97] | supervised | CNN | - | - | 11 datasets | 203,243,090 |
| MiDaS-tf-BEiT-B384-112M [95][‡] | supervised | Transformer | - | - | 10 datasets | 111,531,689 |
| MiDaS-tf-BEiT-L384-344M [95][‡] | supervised | Transformer | - | - | 10 datasets | 344,338,601 |
| MiDaS-tf-BEiT-L512-345M [95][‡] | supervised | Transformer | - | - | 10 datasets | 345,014,441 |
| MiDaS-tf-LeViT-224-51M [95][‡] | supervised | Transformer | - | - | 10 datasets | 50,630,837 |
| MiDaS-tf-NextViT-L384-72M [95][‡] | supervised | Transformer | - | - | 10 datasets | 72,260,713 |
| MiDaS-tf-Swin-L384-213M [95][‡] | supervised | Transformer | - | - | 10 datasets | 213,407,453 |
| MiDaS-tf-Swin2-B384-102M [95][‡] | supervised | Transformer | - | - | 10 datasets | 102,378,913 |
| MiDaS-tf-Swin2-L384-213M [95][‡] | supervised | Transformer | - | - | 10 datasets | 213,411,869 |
| MiDaS-tf-Swin2-T256-42M [95][‡] | supervised | Transformer | - | - | 10 datasets | 41,701,331 |
| MiDaS-hy-H384-123M [74][‡] | supervised | hybrid | - | - | 10 datasets | 123,146,985 |
| MiDaS-hy-H384-123M-K [74][†] | supervised | hybrid | - | ✓ | 10 datasets | 123,146,985 |
| MiDaS-hy-H384-123M-N [74] | supervised | hybrid | ✓ | - | 10 datasets | 123,146,985 |
| MiDaS-cn-L384-344M [74][‡] | supervised | CNN | - | - | 10 datasets | 344,055,465 |
| MiDaSv21-cn-L384-105M [96][‡] | supervised | CNN | - | - | 6 datasets | 105,362,945 |
| MiDaSv21-cn-S256-21M [96][‡] | supervised | CNN | - | - | 6 datasets | 21,320,545 |
| MIM-tf-87M [104] | supervised | Transformer | ✓ | - | - | 87,219,257 |
| MIM-tf-195M [104] | supervised | Transformer | ✓ | - | - | 195,797,493 |
| NeWCRFs-tf-K [71] | supervised | Transformer | - | ✓ | - | 270,444,877 |
| NeWCRFs-tf-N [71] | supervised | Transformer | ✓ | - | - | 270,444,877 |
| Omnidata-tf [94][†] | supervised | Transformer | - | - | 7 datasets | 123,146,985 |
| P2Net-cn-3frames [100][‡] | self-supervised | CNN | ✓ | - | - | 14,840,217 |
| P2Net-cn-5frames [100][‡] | self-supervised | CNN | ✓ | - | - | 14,840,217 |
| PackNet-cn [67][†] | self-supervised | CNN | - | ✓ | - | 128,294,020 |
| SCDepth-v1-cn-D [98][†] | self-supervised | CNN | - | - | DDAD | 14,842,236 |
| SCDepth-v1-cn-K [98][†] | self-supervised | CNN | - | ✓ | - | 14,842,236 |
| SCDepth-v2-cn-N [99][†] | self-supervised | CNN | ✓ | - | - | 14,842,236 |
| SCDepth-v3-cn-B [75][†] | hybrid (disparity) | CNN | - | - | Bonn | 14,842,236 |
| SCDepth-v3-cn-D [75][†] | hybrid (disparity) | CNN | - | - | DDAD | 14,842,236 |

*(Continued)*

**Table 2.** (Continued)

| Models | Strategy | Backbone | Dataset | | | Size |
|---|---|---|---|---|---|---|
| | | | NYU | KITTI | Others | |
| SCDepth-v3-cn-K [75]† | hybrid (disparity) | CNN | - | ✓ | - | 14,842,236 |
| SCDepth-v3-cn-N [75]† | hybrid (disparity) | CNN | ✓ | - | - | 14,842,236 |
| SCDepth-v3-cn-T [75]† | hybrid (disparity) | CNN | - | - | TUM | 14,842,236 |
| StructDepth-cn [105]‡ | self-supervised | CNN | ✓ | - | - | 14,840,217 |
| TrainFlow-cn [106] | self-supervised | CNN | ✓ | - | - | 19,974,541 |
| TransDepth-hy [107] | supervised | hybrid | ✓ | - | - | 247,399,587 |
| ZeroDepth-tf [108] | supervised | Transformer | - | - | 5 datasets | 232,591,380 |
| ZoeDepth-tf-K [73] | supervised | Transformer | - | ✓ | 10 datasets | 344,820,487 |
| ZoeDepth-tf-N [73] | supervised | Transformer | ✓ | - | 10 datasets | 334,816,746 |
| ZoeDepth-tf-NK [73] | supervised | Transformer | ✓ | ✓ | 10 datasets | 346,100,355 |

scenes and real indoor and outdoor images. Finally, DINOv2 [72] leverages an even larger dataset, pretraining its backbone on LVD-142M, a large-scale dataset comprising diverse images, before being fine-tuned on NYU or KITTI.

Each pre-trained model was obtained from publicly available repositories, as detailed in S1 Table. The outputs for the test images were generated using our computational setup (GALLE-RIA UL9C-R49-6 equipped with a GeForce RTX 4090 16GB GPU). In some cases, we made minimal modifications to the publicly released code to ensure compatibility with our computational setup. It is important to emphasize that these modifications did not alter the fundamental architecture or pre-trained weights of the models but were strictly limited to resolving environment-specific compatibility issues and ensuring faithful replication of the intended model behavior as described in the original publications.

The suffixes in the model names indicate specific properties of each model, including the training datasets and model sizes. The suffixes -N/-K/-B/-D/-T represent the datasets used for training: N for NYU [48], K for KITTI [64], B for Bonn [65], D for DDAD [67], T for TUM [66]). The -NK suffix in ZoeDepth [73] indicates that the model was fine-tuned using both the NYU and KITTI datasets. Similarly, the backbone architecture abbreviations are as follows: 'cn' for CNN, 'tf' for Transformer , 'hy' for a hybrid backbone combining CNN and Transformer, and 'df' for Diffusion. The combination of a number followed by M (e.g. –49M, –87M) denotes the number of model parameters in millions. For models like MiDaS [74, 95,96] and SCDepth [75,98,99], multiple versions are available, indicated by a version number suffix (e.g. v3.1, v2). P$^2$Net [100] includes two model configurations, -3frames (one target frame and two source frames) and -5frames (one target frame and four source frames), depending on the number of source frames used when calculating photometric loss.

Additionally, we categorized the model outputs based on the type of depth information they provided: absolute depth (no symbol in Table 2), scale-shift invariant relative depth (indicated by †), and disparity (indicated by ‡). The depth outputs were saved as uint16 format images. For models outputting absolute depth, we stored the depth outputs as raw measurements in millimeters. For models outputting scale-shift invariant relative depth or disparity, we normalized the outputs by linearly scaling the predicted values to the full range of the uint16 format (0 to 65535), mapping the minimum predicted value to 0 and the maximum to 65535.

For DNN models that natively output scale-shift invariant relative depth or disparity maps, we further performed a per-image linear regression to recover pseudo-absolute depth values for comparison with human judgments and ground truth depth. Specifically, for each

image, we prepared two vectorized representations: $I_{out}$, representing the pixel values of the DNN output, and $I_{GT}$, representing the corresponding ground truth pixel values. We then performed linear regression between these two vectors, modeling the relationship as $I_{GT} = s * I_{out} + t$, to estimate the optimal scale $s^*$ and shift $t^*$ parameters that best align the DNN output with the ground truth depth for that image. We applied these image-specific regression coefficients to transform the DNN outputs at the eight points evaluated by human participants into pseudo-absolute depth values using the transformation $z'_{out} = s^* * z_{out} + t^*$. For models outputting disparity, we applied the same regression approach to the disparity maps to obtain $s^*$ and $t^*$. We then converted the disparity values at the eight points using $d'_{out} = s^* * d_{out} + t^*$ and inverted the resulting values to derive pseudo-absolute depth. During this process, we observed negative disparity values for only one or two data points across the entire dataset of 654 images for several DNN models. We clipped these negative values to 0.0 to ensure valid depth estimations. While alternative normalization methods based on median and L1 norm have been proposed [96], we found that these methods tended to produce outlier values in our preliminary experiments. Therefore, we opted for the linear regression approach, which effectively fits the DNN outputs to the ground truth and provide a more robust and stable normalization for our comparative analyses.

## 5.6. Statistical analysis

We calculated 95% confidence intervals (CIs) for the estimates primarily using bootstrapping with random sampling. Specifically, for measures such as the accuracy and inter-individual similarity of human judgments (Table 1), regression analysis based on human judgments data (Fig 3), exponential fittings of human data (Figs 6 and S6), and human-DNN similarity (Figs 4B, 5, 8A, 9A, 10, 11B, and S1), we calculate 95% CIs by replicating random half-splits of human data 1,000 times. As one exception, for per-image fitting coefficients of both humans and DNNs (Figs 4A, 7, 11A, and S7), we obtained 95% CIs by performing bootstrap resampling of the stimuli (1,000 repetitions, with replacement). This approach was necessary because, due to the deterministic nature of DNNs, generating multiple independent outputs from a single DNN model for calculating CIs was not feasible. Another exception is the calculation of 95% CIs for the supplemental dataset analyses. Specifically, we computed 95% CIs of human-DNN similarity related to the supplemental data and original participant data (S4 and S5A Figs) by bootstrap resampling of the original participant data (1,000 repetitions, with replacement). Unlike the main data, we did not use random half-splits in this case because calculating similarity metrics for 2AFC data (i.e., error consistency) requires preserving trial-by-trial agreement patterns, and we aimed for a consistent approach for both relative judgment tasks in the supplemental dataset.

Furthermore, for model rankings comparing accuracy and similarity to human judgments (Figs 8B, 9B, and 11C), we calculated 95% CIs for Spearman correlation coefficients using the pingouin library in Python, instead of using bootstrapping with random sampling. Additionally, we calculated 95% CIs of category-specific ground truth distance (S3A Fig) using "boot_ci" function in the pingouin library.

## Supporting information

**S1 Text. Additional analysis for supplemental dataset and original participant data.**
(PDF)

**S1 Fig. Relationship between DNN accuracy and similarity scores: correlation with ground truth and human judgments.** (A) Scatter plot of scale-shift invariant RMSE versus the Pearson correlation between DNN output and ground truth. (B) Scatter plot of scale-shift invariant RMSE versus the Pearson correlation between DNN outputs and human data.
(TIF)

**S2 Fig. Error pattern similarity among humans and DNNs.** (Left) Heatmap for the absolute data, showing Pearson partial correlations among humans and 31 absolute-value DNNs. (Right) Heatmap for the scale-recovered data, showing Pearson partial correlations among humans and 64 scale-recovered DNNs. In both panels, the top row and leftmost column correspond to human-DNN similarities; DNNs are ordered by descending human similarity.
(TIF)

**S3 Fig. Category-specific analysis for absolute and scale-recovered data.** (A) Per-category RMSE analysis. The top plot shows the average ground truth distance for each semantic category. The plots below show the per-category RMSE for humans (purple squares) and DNNs (dots) for both absolute and scale-recovered data. DNN models are color-coded by their overall similarity to human judgments (measured by point-wise partial correlation). (B) Category-level human similarity for each DNN model. This metric measures the Pearson partial correlation between human and DNN per-category error patterns (from panel A) while controlling for per-category ground truth distance. (C) Relationships between category-level human similarity (from panel B) with overall point-wise human similarity.
(TIF)

**S4 Fig. Results of human-DNN similarity analysis using the individual participant's data without half-split procedure.** (A) Similarity between humans and DNNs based on Pearson partial correlations. (B) Scatter plot showing the relationship between scale-shift invariant RMSE and human similarity. For both absolute (left) and scale-recovered (right) analyses, the inter-human partial correlations were calculated from absolute data, serving as a reference benchmark for human-level consistency. These results are highly consistent with those derived from the half-split procedure (Figs 4B and 5B).
(TIF)

**S5 Fig. Relationship between model accuracy and similarity to human judgments for two measures ('relative' and '2AFC').** (A) Scatter plots illustrating the relationship between scale-shift invariant RMSE and human similarity by two measures. Marker colors indicate the type of training datasets, while marker shapes represent the training strategy used. (B) Spearman correlation coefficients for model rankings based on human similarity across four distinct measures. We analyze both main and supplemental data in this graph using original depth judgments instead of random half-split data.
(TIF)

**S6 Fig. Exponential fitting coefficients for human data and 64 DNNs with scale-recovered data.** The figure consists of three subplots: scale component ($C$), exponent component ($\gamma$), and shift component ($\beta$).
(TIF)

**S7 Fig. Averaged affine coefficients for human data and 64 DNNs with scale-recovered data.** The figure comprises five subplots: scale component ($a_z$), shift component ($b$),

horizontal shear component ($a_x$), vertical shear component ($a_y$), and RMSE for residual error. Error bars represent the 95% confidence intervals derived from random half-split human data.
(TIF)

**S1 Table. Full version of 64 target model list.** This list includes the pre-trained model sources of public repositories in addition to Table 2.
(CSV)

## Author contributions

**Conceptualization:** Yuki Kubota, Taiki Fukiage.

**Data curation:** Yuki Kubota.

**Formal analysis:** Yuki Kubota, Taiki Fukiage.

**Funding acquisition:** Yuki Kubota.

**Investigation:** Yuki Kubota, Taiki Fukiage.

**Methodology:** Yuki Kubota, Taiki Fukiage.

**Project administration:** Yuki Kubota, Taiki Fukiage.

**Resources:** Yuki Kubota.

**Software:** Yuki Kubota, Taiki Fukiage.

**Supervision:** Taiki Fukiage.

**Validation:** Yuki Kubota, Taiki Fukiage.

**Visualization:** Yuki Kubota.

**Writing – original draft:** Yuki Kubota.

**Writing – review & editing:** Yuki Kubota, Taiki Fukiage.

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
