## [Decision Letter · Decision Letter 0]

4 May 2025

PCOMPBIOL-D-25-00627

Human-like monocular depth biases in deep neural networks

PLOS Computational Biology

Dear Dr. Kubota,

Thank you for submitting your manuscript to PLOS Computational Biology. After careful consideration, we feel that it has merit but does not fully meet PLOS Computational Biology's publication criteria as it currently stands. Therefore, we invite you to submit a revised version of the manuscript that addresses the points raised during the review process.

Please submit your revised manuscript within 60 days Jul 04 2025 11:59PM. If you will need more time than this to complete your revisions, please reply to this message or contact the journal office at ploscompbiol@plos.org. Please include the following items when submitting your revised manuscript:

We look forward to receiving your revised manuscript.

Kind regards,

Jian Liu

Academic Editor

PLOS Computational Biology

Marieke van Vugt

Section Editor

PLOS Computational Biology

**Additional Editor Comments :**

The manuscript presents some interesting work. However, there are a few concerns and suggestions to improve it. Please review the manuscript and consider addressing these comments.

**Journal Requirements:**

1) Please ensure that the CRediT author contributions listed for every co-author are completed accurately and in full.At this stage, the following Authors/Authors require contributions: Yuki Kubota, and Taiki Fukiage. Please ensure that the full contributions of each author are acknowledged in the "Add/Edit/Remove Authors" section of our submission form.The list of CRediT author contributions may be found here: https://journals.plos.org/ploscompbiol/s/authorship#loc-author-contributions 2) We ask that a manuscript source file is provided at Revision. Please upload your manuscript file as a .doc, .docx, .rtf or .tex. If you are providing a .tex file, please upload it under the item type u2018LaTeX Source Fileu2019 and leave your .pdf version as the item type u2018Manuscriptu2019. 3) Thank you for including an Ethics Statement for your study. Please include:i) A statement that formal consent was obtained (must state whether verbal/written) OR the reason consent was not obtained (e.g. anonymity). NOTE: If child participants, the statement must declare that formal consent was obtained from the parent/guardian.]. 4) Please upload all main figures as separate Figure files in .tif or .eps format. For more information about how to convert and format your figure files please see our guidelines: https://journals.plos.org/ploscompbiol/s/figures 5) We have noticed that you have uploaded Supporting Information files, but you have not included a complete list of legends. Please add a full list of legends for your Supporting Information files (analysis.zip) after the references list. 6) We notice that your supplementary Figures, and information are included in the manuscript file. Please remove them and upload them with the file type 'Supporting Information'. Please ensure that each Supporting Information file has a legend listed in the manuscript after the references list. 7) Some material included in your submission may be copyrighted. According to PLOSu2019s copyright policy, authors who use figures or other material (e.g., graphics, clipart, maps) from another author or copyright holder must demonstrate or obtain permission to publish this material under the Creative Commons Attribution 4.0 International (CC BY 4.0) License used by PLOS journals. Please closely review the details of PLOSu2019s copyright requirements here: PLOS Licenses and Copyright. If you need to request permissions from a copyright holder, you may use PLOS's Copyright Content Permission form.Please respond directly to this email and provide any known details concerning your material's license terms and permissions required for reuse, even if you have not yet obtained copyright permissions or are unsure of your material's copyright compatibility. Once you have responded and addressed all other outstanding technical requirements, you may resubmit your manuscript within Editorial Manager. Potential Copyright Issues:i) Please confirm (a) that you are the photographer of 1, and 12, or (b) provide written permission from the photographer to publish the photo(s) under our CC BY 4.0 license.  8) Thank you for stating "Human and model data, and data analysis code are available on Github (https://github.com/yuki-kubota-95/human-like-MDE)." This link reaches a 404 error page. Please amend this to a new link or provide further details to locate the data. 

**Reviewers' comments:**

Reviewer's Responses to Questions

Reviewer #1: This study investigates how humans and deep neural networks (DNNs) perceive depth from 2D images, revealing that both exhibit systematic distortions such as depth compression and vertical biases. By analyzing a novel human-annotated dataset using affine decomposition, the authors show that more accurate DNNs partially mirror human error patterns, suggesting these biases may reflect ecologically efficient strategies. The work highlights the value of examining error structures beyond raw accuracy to better understand human and model depth perception.

The paper is well-written, well-organized, and easy to follow. The use of absolute depth judgments sets it apart from existing studies, and the experimental design is thoughtful and adds meaningful value to the community. The analysis of depth biases in both humans and models is particularly insightful.

However, I have several concerns regarding the evaluation metrics, experimental design, and missing comparisons with recent state-of-the-art models. Addressing these issues would significantly strengthen the paper.

Error Patterns and Adversarial Stimuli

1. If the authors can identify systematic error patterns in both human and model predictions, they could potentially design adversarial stimuli that exploit these patterns to elicit mistakes. This direction would be a valuable avenue for future work and should be discussed.

Human-Human Consistency as a Baseline

2. What is the depth estimation consistency across different human observers? Quantifying this would provide a strong baseline for evaluating human-machine agreement. Similarly, consistency across different machine models should also be reported.

RMSE for Humans

3. Why is RMSE reported only for machine predictions? It would be informative to compute the same metric for human estimates to allow for a more direct and fair comparison.

Depth Comparisons: Human vs Machine

4. In Figure 2, human depth estimates are compared to ground truth. Why isn’t the same comparison made for machine predictions? Instead of comparing RMSE values indirectly, it would be more informative to directly compare machine-predicted depths to human estimates.

Correlation Values

5. Building on point 4, what are the correlation values between machine-predicted depths and ground truth? Additionally, what is the correlation between machine estimates and human estimates?

Figure Legends

6. In Figures 9 and 11 (and possibly others), the light pink bars represent human performance. This should be clearly indicated in the figure legends, which are currently missing this information.

Missing Related Work on Human-Machine Comparison

7. Important studies on human-machine comparisons in other cognitive domains such as working memory and visual search are missing from the related work. Please consider including and discussing relevant works such as:

[a] https://arxiv.org/abs/2307.10768

[b] https://journals.plos.org/ploscompbiol/article?id=10.1371/journal.pcbi.1010654

Recent Advances in Depth Inference Models

8. Recent models like Neural Radiance Fields (NeRF) and 3D Gaussian Splatting can infer depth from images. Can the authors compare these models to human performance, or at least discuss their relevance in the related works section?

Object Semantics in Depth Estimation

9. Object-level semantics can aid depth estimation—e.g., knowing the typical size of a sofa provides depth cues. Since the dataset includes ground truth segmentation masks, can the authors analyze category-specific depth estimation or explore whether there are any categorical biases?

Data Availability Statement

10. The manuscript is missing a data availability statement. Please include one to support reproducibility.

Reviewer #2: This work offers a novel perspective on comparing human depth perception with that of deep neural network (DNN)–based computer vision models. Through extensive statistical analyses and data comparisons, the authors explore both the similarities and differences in performance and attentional focus between humans and DNNs.

However, the experimental setup lacks assessment and documentation of individual participants’ habits and capabilities—for example, there is no record of ocular dominance (preferred or dominant eye). The authors attempt to mitigate individual behavioral biases by averaging across a random half‐split of the human data, but the methodology and rationale for this “random half‐split” procedure are not described in sufficient detail.

In addition, the manuscript’s stated motivation feels somewhat misaligned. The abstract frames the study as an investigation into distortions and biases in human 2D vision assisted by DNN, yet the Discussion shifts focus to the comparison of DNNs and human visual distortions. It might be clearer—and more impactful—to reposition the work as establishing a benchmark comparison between human observers and various state-of-the-art depth-estimation models.

Moreover, while the paper employs monocular depth-estimation DNNs, it is not specified whether the human experiments were conducted monocularly or binocularly. This mismatch in input conditions could influence the findings and should be clarified or controlled.

Finally, I recommend refining the layout for greater visual clarity: adjust figure placements and font sizes to ensure proper alignment and aesthetics, and relocate or resize some appendix images that currently interfere with the reference list.

**Have the authors made all data and (if applicable) computational code underlying the findings in their manuscript fully available?**

Reviewer #1: **No: **As far as I can tell, the zip folder only contains the post-processed results and the plots of the results. The models, the evaluation code, and the code for conducting human behavioral experiments are missing.

PLOS authors have the option to publish the peer review history of their article (what does this mean?). If published, this will include your full peer review and any attached files.

Reviewer #1: **Yes: **Mengmi Zhang

Reviewer #2: No

**Figure resubmission:**
---

## [Decision Letter · Decision Letter 1]

15 Jul 2025

Dear Dr. Kubota,

We are pleased to inform you that your manuscript 'Human-like monocular depth biases in deep neural networks' has been provisionally accepted for publication in PLOS Computational Biology.

Best regards,

Jian Liu

Academic Editor

PLOS Computational Biology

Marieke van Vugt

Section Editor

PLOS Computational Biology

Reviewer's Responses to Questions

**Comments to the Authors:**

Reviewer #1: I appreciate the responses from the authors and the edits made in the revised version of the manuscript.

The rebuttal has addressed all my initial concerns.

Reviewer #2: The author has incorporated ample supporting information and carried out scientifically precise revisions in accordance with the reviewers’ comments. Through these additions and modifications, the manuscript now demonstrates a rigorous, scientifically sound experimental design, robust reproducibility, and genuine novelty, fully satisfying the essential criteria for publication.

**Have the authors made all data and (if applicable) computational code underlying the findings in their manuscript fully available?**

Reviewer #1: Yes

Reviewer #2: Yes

PLOS authors have the option to publish the peer review history of their article (what does this mean?). If published, this will include your full peer review and any attached files.

Reviewer #1: **Yes: **Mengmi Zhang

Reviewer #2: **Yes: **Shixiao Wang

---

## [Editor Report · Acceptance letter]

PCOMPBIOL-D-25-00627R1

Human-like monocular depth biases in deep neural networks

Dear Dr Kubota,

I am pleased to inform you that your manuscript has been formally accepted for publication in PLOS Computational Biology. Your manuscript is now with our production department and you will be notified of the publication date in due course.

With kind regards,

Benedek Toth
